# Towards domain-invariant Self-Supervised Learning with Batch Styles Standardization

**Marin Scalbert & Maria Vakalopoulou**
MICS, CentraleSupélec, Université Paris-Saclay
Gif-sur-Yvette, France
`{name.surname}@centralesupelec.fr`

**Florent Couzinié-Devy**
VitaDX
Paris, France
`f.couzinie-devy@vitadx.com`

## Abstract

In Self-Supervised Learning (SSL), models are typically pretrained, fine-tuned, and evaluated on the same domains. However, they tend to perform poorly when evaluated on unseen domains, a challenge that Unsupervised Domain Generalization (UDG) seeks to address. Current UDG methods rely on domain labels, which are often challenging to collect, and domain-specific architectures that lack scalability when confronted with numerous domains, making the current methodology impractical and rigid. Inspired by contrastive-based UDG methods that mitigate spurious correlations by restricting comparisons to examples from the same domain, we hypothesize that eliminating style variability within a batch could provide a more convenient and flexible way to reduce spurious correlations without requiring domain labels. To verify this hypothesis, we introduce Batch Styles Standardization (BSS), a relatively simple yet powerful Fourier-based method to standardize the style of images in a batch specifically designed for integration with SSL methods to tackle UDG. Combining BSS with existing SSL methods offers serious advantages over prior UDG methods: (1) It eliminates the need for domain labels or domain-specific network components to enhance domain-invariance in SSL representations, and (2) offers flexibility as BSS can be seamlessly integrated with diverse contrastive-based but also non-contrastive-based SSL methods. Experiments on several UDG datasets demonstrate that it significantly improves downstream task performances on unseen domains, often outperforming or rivaling UDG methods. Finally, this work clarifies the underlying mechanisms contributing to BSS's effectiveness in improving domain-invariance in SSL representations and performances on unseen domains. Implementations of the extended SSL methods and BSS are provided at this url.

## 1 Introduction

**Motivations.** In recent years, Self-Supervised Learning (SSL), has seen significant growth and success (Chen et al., 2020a; Grill et al., 2020; Caron et al., 2020; 2021; Assran et al., 2022; Bardes et al., 2021; He et al., 2022). However, SSL generally assumes that pretraining, fine-tuning and testing data come from the same domains, an assumption which does not hold true in practice and thereby limits its real-life applications. The distribution shifts between pretraining/fine-tuning domains (sources domains) and testing domains (targets domains) usually lead to poor generalization on testing domains.

Unsupervised Domain Generalization (UDG) (Zhang et al., 2022), aims to tackle this issue by evaluating how well fine-tuned SSL models generalize to unseen target domains. In UDG, models are first pretrained on unlabeled data, fine-tuned on labeled data and finally evaluated on data from unseen domains. This work focuses on the *all-correlated* UDG setting which is the most standard and studied one (Zhang et al., 2022; Harary et al., 2022; Yang et al., 2022b). In this setting, unlabeled and labeled data come from the same source domains, testing data come from unseen target domains, while all cover the same classes.

Current UDG methods suffer from the same drawbacks: (1) They require domain labels to reinforce domain-invariance in SSL representations, while in practice these labels may be challenging

to obtain or even unavailable and (2) they all rely on domain-specific architectures, such as domain-specific negative queues or domain-specific decoders, that lack scalability when confronted to numerous domains. These limitations highlight the need for more practical and flexible UDG methods.

Taking inspiration from contrastive-based UDG methods that reduce spurious correlations by restraining comparisons to examples from the same domain, we believe that removing style variability within a batch through style standardization may provide a more practical and flexible way to mitigate spurious correlations and achieve domain-invariant SSL representations without requiring any domain labels.

**Contributions.** To investigate the effectiveness of style standardization in mitigating spurious correlations within SSL representations with the aim of proposing more convenient and flexible UDG approaches, we introduce Batch Styles Standardization (BSS). BSS is a simple yet powerful Fourier-based method for standardizing image styles within a batch purposefully designed for integration with existing SSL methods to reinforce domain-invariance. Style standardization is performed by transferring the style of a randomly selected image to all images in the batch.

Integrated with existing SSL methods, it confers significant advantages over prior UDG works: (1) it reinforces domain-invariance without requiring any domain labels or domain-specific architecture, and (2) it offers simplicity and flexibility, integrating easily with contrastive-based (SimCLR (Chen et al., 2020a), SWaV (Caron et al., 2020)) but also non-contrastive-based SSL methods (MSN (Assran et al., 2022)).

Experiments conducted on UDG datasets indicate that BSS combined with the different SSL methods yields significant performance gains on unseen domains while outperforming or competing with established UDG methods. Finally, extensive experiments have been conducted to clarify the underlying mechanisms driving BSS's effectiveness in enhancing domain-invariance in SSL representations and performances on unseen domains.

## 2 RELATED WORKS

**Domain Generalization.** DG aims to learn a model from multiple source domains with distinct distributions to generalize well to unseen target domains. Former DG methods have focused on aligning source features distributions using a large panel of techniques (Ganin et al., 2016; Kang et al., 2019; Li et al., 2018a;b; Peng et al., 2019; Scalbert et al., 2021; Zhao et al., 2020). Recently, the trend has shifted towards improving cross-domain generalization by refining data augmentation strategies. These strategies can be applied at either the image level (Scalbert et al., 2022; Xu et al., 2021; Yang & Soatto, 2020; Zhou et al., 2020a;b) or the feature representation level (Kang et al., 2022; Li et al., 2021; Zhou et al., 2021), and can be non-parametric (Xu et al., 2021; Zhou et al., 2021), trained adversarially during the DG task (Hoffman et al., 2018; Kang et al., 2022; Zhou et al., 2020a;b), or pretrained beforehand on source domains (Scalbert et al., 2022). Among these methods, Fourier-based Augmentations (FA) (Xu et al., 2021; Yang & Soatto, 2020) stand out as a simple and promising approach to instill domain-invariance into the representations and, thereby, enhance generalization. In this work, the proposed BSS extends FA's style transfer ability to standardize the style of images within a batch so as to strengthen domain-invariance in SSL methods.

**Self-Supervised Learning.** SSL has gained a lot of attention for its ability to efficiently pretrain models on abundant unlabeled data and subsequently fine-tune them for downstream tasks with limited labeled data. Contrastive and non-contrastive-based methods have emerged as successful approaches. The former focuses on making representations of similar examples (positives) closer while pushing apart representations of dissimilar examples (negatives). Similar examples are usually built by generating several augmented views of the same image. These methods operate either at the instance-level (Chen et al., 2020a;c; Hu et al., 2021) or cluster-level (Caron et al., 2018; 2020). Given their reliance on a large number of negatives, non-contrastive-based methods have attempted to eliminate the use of negative examples but require additional tricks to avoid collapse (Grill et al., 2020; Chen & He, 2021; Caron et al., 2021). In this work, harnessing FA and the proposed BSS, we extend both contrastive-based (SimCLR, SWaV) and non-contrastive-based (MSN) methods to strengthen domain-invariance and address UDG.

**Unsupervised Domain Generalization.** Contrastive-based UDG methods (DARLING (Zhang et al., 2022), BrAD (Harary et al., 2022)) improve domain-invariance by ensuring that positive

and negative examples share the same domain. This constraint mitigates spurious correlations within SSL representations when repelling negative examples from positive ones. To respect this constraint, DARLING exploits domain-specific adversarial negative queues while BrAD maintains domain-specific negative queues containing past representations of a momentum encoder. Additionally, BrAD learns image-to-image mappings from the different domains to a shared space and compares representations of raw and projected images. As an alternative, DiMAE (Yang et al., 2022b) and CycleMAE (Yang et al., 2022a) rely on Masked Auto-Encoder (He et al., 2022) (MAE) with domain-specific decoders to solve a cross-domain reconstruction task. However, these methods rely on domain labels and complex domain-specific architectures, limiting scalability and adaptability. Inspired by UDG contrastive methods, we propose removing style variability within a batch and without domain labels to reduce spurious correlations in SSL methods resulting in simpler and more flexible UDG approaches.

## 3 METHOD

### 3.1 PROBLEM FORMULATION

In the *all-correlated* UDG setting, an unlabeled dataset, a labeled dataset and a test dataset are provided. Unlabeled and labeled training data are drawn from the same source domains $\mathcal{D}_S$ while testing data are drawn from unseen target domains $\mathcal{D}_T$. All data share the same class labels space $\mathcal{Y}$. The goal of UDG is to pretrain a model on the unlabeled data, fine-tune it on the labeled data and achieve good generalization on the test dataset.

### 3.2 BATCH STYLES STANDARDIZATION

#### 3.2.1 PRELIMINARIES ON FOURIER-BASED AUGMENTATIONS

Fourier-based Augmentations (Xu et al., 2021; Yang & Soatto, 2020) are motivated by a property of the Fourier transform: phase components tend to retain semantic information while amplitude components the style information such as intensity and textures. Therefore, to make the network prioritize semantics over style, FA randomly alters the amplitudes of images during training.

More formally, given an image $\boldsymbol{X} \in \mathbb{R}^{H \times W}$, its Fourier transform $\mathcal{F}(\boldsymbol{X})$ along with the corresponding amplitude $\mathcal{A}(\boldsymbol{X})$ and phase $\mathcal{P}(\boldsymbol{X})$ are computed as follows:

$$\mathcal{F}(\boldsymbol{X})(u,v) = \sum_{h=1}^{H} \sum_{w=1}^{W} X_{h,w} e^{-i2\pi \left( \frac{h}{H} u + \frac{w}{W} v \right)} \tag{1}$$

$$= \mathcal{A}(\boldsymbol{X})(u,v) e^{-i\mathcal{P}(\boldsymbol{X})(u,v)} \tag{2}$$

$$\text{where } \mathcal{A}(\boldsymbol{X}) = \sqrt{\text{Re}\left(\mathcal{F}(\boldsymbol{X})\right)^2 + \text{Im}\left(\mathcal{F}(\boldsymbol{X})\right)^2} \text{ and } \mathcal{P}(\boldsymbol{X}) = \arctan\left( \frac{\text{Im}\left(\mathcal{F}(\boldsymbol{X})\right)}{\text{Re}\left(\mathcal{F}(\boldsymbol{X})\right)} \right) \tag{3}$$

The amplitude $\mathcal{A}(\boldsymbol{X})$ is then altered by substituting its low-frequency components with those of a randomly selected image $\mathcal{A}(\boldsymbol{X}')$ resulting in the altered amplitude $\hat{\mathcal{A}}(\boldsymbol{X})$:

$$\hat{\mathcal{A}}(\boldsymbol{X})(u,v) = \begin{cases} \mathcal{A}(\boldsymbol{X}')(u,v) & \text{if } u \leq r * H \text{ and } v \leq r * W, \\ \mathcal{A}(\boldsymbol{X})(u,v) & \text{if } u > r * H \text{ and } v > r * W, \end{cases} \tag{4}$$

The strength of the augmentation is controlled by the hyperparameter $r \sim U(r_{min}, r_{max})$ representing the ratio between the substituted amplitude area and the entire amplitude area, where $r_{min}$ and $r_{max}$ stand for the minimum and maximum possible ratios. Finally, an augmented image $\hat{\boldsymbol{X}}$ with the same content as the original image $\boldsymbol{X}$ and style as the randomly chosen image $\boldsymbol{X}'$ can be built by applying the inverse Fourier transform $\mathcal{F}^{-1}$ onto the altered amplitude $\hat{\mathcal{A}}(\boldsymbol{X})$ and unmodified phase $\mathcal{P}(\boldsymbol{X})$:

$$\hat{\boldsymbol{X}} = \mathcal{F}^{-1}\left( \hat{\mathcal{A}}(\boldsymbol{X}) e^{-i\mathcal{P}(\boldsymbol{X})} \right) \tag{5}$$

### 3.2.2 EXTEND FOURIER-BASED AUGMENTATIONS FOR BATCH STYLES STANDARDIZATION

Drawing inspiration from contrastive-based UDG methods, we believe that removing style variability within a batch might reduce spurious correlations in SSL methods without requiring domain labels resulting in simpler and more flexible UDG approaches. Since FA possess a style transfer-like ability, they can be extended to perform styles standardization/harmonization. Concretely, it can be accomplished by transferring the style of a randomly chosen image within the batch to all other images in that batch. Hence, the proposed method is referred to as Batch Styles Standardization.

The process of applying BSS is illustrated on Figure 1a. Specifically, given a batch of images and their corresponding Fourier transforms, we manipulate the different amplitudes by substituting their low-frequency components with those of a single randomly chosen image. Finally, after applying the inverse Fourier transform to the different modified Fourier transforms, the style of the randomly chosen image is transferred to all images, effectively standardizing/harmonizing the style. A pseudo-code along with a PyTorch implementation of BSS are provided in Appendix A.

To highlight batch-level differences between standard FA and the proposed BSS, we display in Figure 1b and Figure 1c, a $N \times V$ grid of augmented images generated by applying FA or BSS $V$ times on a batch of $N$ images. For a specific view index (column index), it is clear that augmented images produced by FA exhibit different styles whereas in the case of BSS, a unique style prevails. It is important to notice that standardized images can undergo independent geometric augmentations, but color augmentations must be batch-wise to preserve the unique style.

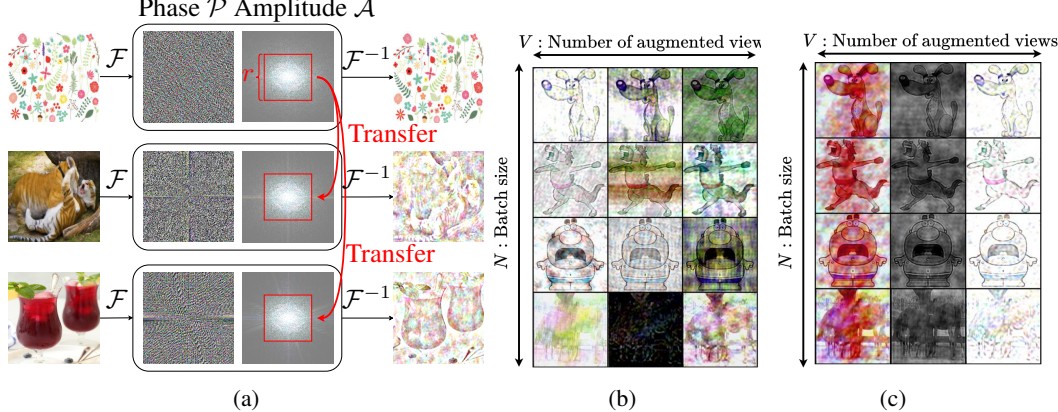

Figure 1: (a) BSS: Fourier Transform $\mathcal{F}$ is applied on all batch images then low-frequency components of the amplitudes $\mathcal{A}$ (determined by the areas ratio $r$) are replaced by those of a randomly chosen image (the first one in this case). Finally, inverse Fourier transform $\mathcal{F}^{-1}$ is applied to the altered Fourier transforms to build images with standardized styles. (b) Augmented images with standard independent FA. (c) Augmented images with BSS.

As opposed to UDG methods exploiting domain labels to restrict comparisons to examples from the same domain, standardizing the style of examples using a random style simulates as if they were drawn from the same "pseudo-domain", thereby eliminating the need for domain labels. Moreover, BSS's seamless integration into existing SSL methods removes the need for UDG domain-specific components, such as domain-specific negative queues (DARLING, BRaD) or domain-specific decoders (DiMAE). Collectively, these characteristics position BSS as a simpler and more versatile solution to enhance domain-invariance within SSL methods and address UDG.

### 3.3 HOW TO INTEGRATE BATCH STYLES STANDARDIZATION INTO SSL METHODS?

Both contrastive and non-contrastive methods aim to distribute batch examples over the embedding space. This distribution can be driven by explicit contrastive loss (SimCLR) or methods preventing representation collapse like the Sinkhorn-Knopp algorithm (SWaV, MSN), centering (DINO), or variance regularization (VicReg). However, when dealing with diverse domains/styles within a batch, distributing examples may unintentionally group them by domains/styles, resulting in spuri-

ous correlations. To mitigate this, we propose applying BSS to examples undergoing this distribution. Removing style information through standardization should encourage the distribution to focus more on example semantics. In the following sections, we extend three existing SSL methods (SimCLR, SWaV and MSN) and detail where and how BSS should be integrated. For further technical details about the regular SSL methods, readers may refer to Appendix B or the original papers.

**SIMCLR** aims to bring representations of several views of the same image closer (positives) while repelling all other images representations (negatives). In standard SimCLR, each image in a batch of $N$ images is augmented $V$ times resulting in $V$ positive examples. However, this can result in positive examples with domains/styles that differ from those of the negatives (see Figure 1b), risking unintentional exploitation of this style/domain discrepancy to solve the contrastive task and causing spurious correlations. To address this, we suggest independently applying BSS $V$ times to the initial batch, ensuring that positive and negative examples share the same styles (see Figure 1c).

**SWAV** computes representations of several views of the same image and clusters them using an online algorithm. Given that representations should capture similar information, SWaV enforces consistency between representations and cluster assignments produced from different views. To obtain these cluster assignments, the Sinkhorn-Klopp (SK) algorithm (Cuturi, 2013) is performed on the representations. Concretely, SK solves an optimal transport problem whose constraints are to assign representations to the most similar centroids/prototypes while keeping a uniform assignment distribution over centroids/prototypes. However, if several domains/styles are present within views subject to SK, there is a risk of assigning and grouping the corresponding representations using domain/style information resulting in spurious correlations. To address this, we propose to standardize the style of views subject to SK using BSS. In practice, SWaV employs a multi-crop strategy, generating 2 global views (large crops) and $V$ local views (small crops) for each image. In this setting, cluster assignments are computed only from the global views while representations are derived from all views. Therefore, we suggest applying BSS only on the global views and augmenting the local views using FA. As this results in two batches of global views, each with its own style, SK is performed on each batch separately.

**MSN** aims to match the representations of masked views of the same image with that of an unmasked view. To derive a view's representation, MSN computes similarities between its embedding and a set of learnable cluster centroids/prototypes and subsequently transforms them into a probability distribution. Since direct matching between masked and unmasked views' representations can lead to representation collapse, MSN simultaneously optimizes a cross-entropy term along with an entropy regularization term on the mean representation of the masked views. This regularization term encourages the model to use the entire set of centroids/prototypes. Additionally, MSN employs SK on the representations of the unmasked views to avoid tuning the hyperparameter weighting the entropy regularization term. Similarly to SWaV, if several domains/styles are present within unmasked views, assigning the corresponding representations using SK may group examples using domain/style information. To address this, we propose to standardize the style of unmasked views using BSS while we recommend augmenting masked views using FA.

## 4 RESULTS

### 4.1 DATASETS

To evaluate the extended SSL methods, experiments were conducted on 3 datasets commonly used for benchmarking DG / UDG methods, namely **PACS**, **DomainNet** and **Camelyon17 WILDS**.

**PACS** (Li et al., 2017) contains 4 domains (*photo*, *art painting*, *cartoon*, *sketch*) and 7 classes. **DomainNet** (Peng et al., 2019) contains 6 different domains (*clipart*, *infograph*, *quickdraw*, *painting*, *real* and *sketch*) and covers 345 classes. Following prior UDG works (Harary et al., 2022; Yang et al., 2022b; Zhang et al., 2022), a subset of DomainNet including 20 classes out of the 345 available classes is considered. **Camelyon17 WILDS** (Koh et al., 2021) includes images covering 2 classes (tumor, no tumor) from 5 domains (hospitals). It is split into `train`, `val`, and `test` subsets comprising respectively 3, 1, and 1 distinct domains.

Table 1: UDG performances on PACS. Best methods are highlighted in **bold**.

| Methods | Label Fraction: 1% — Target domain | | | | | Label Fraction: 5% — Target domain | | | | |
|---|---|---|---|---|---|---|---|---|---|---|
| | *photo* | *art* | *cartoon* | *sketch* | *avg.* | *photo* | *art* | *cartoon* | *sketch* | *avg.* |
| ERM | 10.90 | 11.21 | 14.33 | 18.83 | 13.82 | 14.15 | 18.67 | 13.37 | 18.34 | 16.13 |
| MoCo V2 | 22.97 | 15.58 | 23.65 | 25.27 | 21.87 | 37.39 | 25.57 | 28.11 | 31.16 | 30.56 |
| SimCLR V2 | 30.94 | 17.43 | 30.16 | 25.20 | 25.93 | 54.67 | 35.92 | 35.31 | 36.84 | 40.68 |
| BYOL | 11.20 | 14.53 | 16.21 | 10.01 | 12.99 | 26.55 | 17.79 | 21.87 | 19.65 | 21.46 |
| AdCo | 26.13 | 17.11 | 22.96 | 23.37 | 22.39 | 37.65 | 28.21 | 28.52 | 30.35 | 31.18 |
| MAE | 30.72 | 23.54 | 20.78 | 24.52 | 24.89 | 32.69 | 24.61 | 27.35 | 30.44 | 28.77 |
| DARLING | 27.78 | 19.82 | 27.51 | 29.54 | 26.16 | 44.61 | 39.25 | 36.41 | 36.53 | 39.20 |
| DiMAE[*] | 48.86 | 31.73 | 25.83 | 32.50 | 34.73 | 50.00 | 41.25 | 34.40 | 38.00 | 40.91 |
| BrAD[*] | **61.81** | 33.57 | 43.47 | 36.37 | 43.80 | **65.22** | 41.35 | 50.88 | 50.68 | 52.03 |
| CycleMAE[*] | 52.63 | 36.25 | 35.53 | 34.85 | 39.82 | 63.24 | 39.96 | 42.15 | 36.35 | 45.43 |
| SimCLR w/ FA | 41.00 | 37.94 | 45.38 | 43.47 | 41.95 | 57.17 | 44.78 | 50.16 | 55.32 | 51.85 |
| SimCLR w/ BSS | 43.31 (↑ 2.31) | **38.96** (↑ 1.02) | **48.61** (↑ 3.23) | **48.76** (↑ 5.29) | 44.91 (↑ 2.96) | 58.16 (↑ 0.99) | **46.37** (↑ 1.59) | **55.69** (↑ 5.53) | **65.63** (↑ 10.04) | **56.40** (↑ 4.55) |
| SWaV w/ FA | 36.15 | 32.93 | 36.63 | 27.37 | 33.27 | 41.64 | 40.95 | 48.51 | 45.32 | 44.10 |
| SWaV w/ BSS | 39.74 (↑ 3.59) | 35.82 (↑ 2.89) | 42.59 (↑ 5.96) | 36.12 (↑ 8.75) | 38.57 (↑ 5.3) | 50.58 (↑ 8.94) | 43.00 (↑ 2.05) | 53.81 (↑ 5.3) | 52.61 (↑ 7.29) | 50.00 (↑ 5.9) |
| | Label Fraction: 10% — Target domain | | | | | Label Fraction: 100% — Target domain | | | | |
| Methods | *photo* | *art* | *cartoon* | *sketch* | *avg.* | *photo* | *art* | *cartoon* | *sketch* | *avg.* |
| ERM | 16.27 | 16.62 | 18.40 | 12.01 | 15.82 | 43.29 | 24.27 | 32.62 | 20.84 | 30.26 |
| MoCo V2 | 44.19 | 25.85 | 35.53 | 24.97 | 32.64 | 59.86 | 28.58 | 48.89 | 34.79 | 43.03 |
| SimCLR V2 | 54.65 | 37.65 | 46.00 | 28.25 | 41.64 | 67.45 | 43.60 | 54.48 | 34.73 | 50.06 |
| BYOL | 27.01 | 25.94 | 20.98 | 19.69 | 23.40 | 41.42 | 23.73 | 30.02 | 18.78 | 28.49 |
| AdCo | 46.51 | 30.31 | 31.45 | 22.96 | 32.81 | 58.59 | 29.81 | 50.19 | 30.45 | 42.26 |
| MAE | 35.89 | 25.59 | 33.28 | 32.39 | 31.79 | 36.84 | 25.24 | 32.25 | 34.45 | 32.20 |
| DARLING | 53.37 | 39.91 | 46.41 | 30.17 | 42.46 | 68.86 | 41.53 | 56.89 | 37.51 | 51.20 |
| DiMAE[*] | 77.87 | 59.77 | 57.72 | 39.25 | 58.65 | 78.99 | 63.23 | 59.44 | 55.89 | 64.39 |
| BrAD[*] | 72.17 | 44.20 | 50.01 | 55.66 | 55.51 | ✗ | ✗ | ✗ | ✗ | ✗ |
| CycleMAE[*] | **85.94** | **67.93** | 59.34 | 38.25 | **62.87** | **90.72** | **75.34** | **69.33** | 50.24 | **71.41** |
| SimCLR w/ FA | 62.67 | 49.92 | 54.79 | 58.32 | 56.43 | 78.36 | 59.41 | 65.16 | 63.59 | 66.63 |
| SimCLR w/ BSS | 63.29 (↑ 0.62) | 51.37 (↑ 1.45) | **59.43** (↑ 4.64) | **66.09** (↑ 7.77) | 60.04 (↑ 3.61) | 79.50 (↑ 1.14) | 62.73 (↑ 3.32) | 65.67 (↑ 0.51) | **73.02** (↑ 9.43) | 70.23 (↑ 3.6) |
| SWaV w/ FA | 46.27 | 44.68 | 50.27 | 50.02 | 47.81 | 77.50 | 57.49 | 64.32 | 66.08 | 66.35 |
| SWaV w/ BSS | 57.82 (↑ 11.55) | 45.91 (↑ 1.23) | 53.65 (↑ 3.38) | 55.67 (↑ 5.65) | 53.27 (↑ 5.46) | 78.62 (↑ 1.12) | 59.65 (↑ 2.16) | 65.40 (↑ 1.08) | 67.80 (↑ 1.72) | 67.87 (↑ 1.52) |

[*] Uses Imagenet transfer learning.

## 4.2 EXPERIMENTAL SETUP

Following the standard UDG evaluation protocol (Zhang et al., 2022), models were pretrained on source data in an unsupervised way, fine-tuned on a fraction of the source data and finally evaluated on the target data. For the pretraining step, all our models were trained using FA or BSS without Imagenet (Deng et al., 2009) transfer learning, except on **DomainNet** to allow fair comparisons with prior UDG works. In Appendix D.1, the choice of using transfer learning within a DG/UDG context is further discussed while additional experiments on PACS reveal that SSL methods tend to benefit unfairly from ImageNet transfer learning. For the fine-tuning step, following BrAD, on **PACS** and **DomainNet**, all the models were fine-tuned via linear probing except when considering the entire PACS dataset where full fine-tuning was performed like DARLING and DiMAE. On **Camelyon17 WILDS**, linear probing was performed for each fraction of labeled data. Pretraining and fine-tuning implementation details are provided in Appendix C.

## 4.3 EXPERIMENTAL RESULTS

In the following experiments, the proposed extended SSL methods are compared to regular SSL methods (MoCo V2 (Chen et al., 2020c), SimCLR V2 (Chen et al., 2020b), BYOL (Grill et al., 2020), AdCo (Hu et al., 2021), MAE (He et al., 2022)) and UDG methods (DARLING, DiMAE, BRaD, CycleMAE). For Camelyon17 WILDS, the extended SSL methods are compared to reimplemented UDG methods (DARLING, DiMAE) but also to the Semi-Supervised Learning method FixMatch (Sohn et al., 2020) and SSL method SWaV trained with additional data from the target.

**PACS.** For each combination of (sources, target) domains, each fraction of labeled data and each of our SSL models, averaged accuracy over 3 independent runs are reported on Table 1. Compared to FA, integrating BSS to SimCLR or SWaV, significantly improves the overall accuracy (avg.): SimCLR → $(+2.96\%, +4.55\%, +3.61\%, +3.6\%)$; SWaV → $(+5.3\%, +5.9\%, +5.46\%, +1.52\%)$ for the fractions of labeled data $1\%, 5\%, 10\%$ and $100\%$, respectively. Extended SSL methods with BSS outperform most of the time other methods, except for the target domain *photo* (BrAD, DiMAE, CycleMAE) or in the $10\%$ (DiMAE) and $100\%$ (DiMAE, CycleMAE) labeled data settings. For the target domain photo, one possible explanation is that other methods benefit from transfer learning on ImageNet while for the $10\%$ and $100\%$ labeled data settings, DiMAE and CycleMAE consider different experimental settings (ViT-base architecture, full fine-tuning on $10\%$.)

**DomainNet.** Following prior UDG methods, *painting*, *real* and *sketch* were selected as source domains and others as target domains. The reversed domains combination was also considered. For these two combinations, for each fraction of labeled data (1%, 5% and 10%) and each of our SSL models, we report on Table 2, the accuracy on each target domain, the per-domain averaged accuracy and the overall accuracy. The presented results are averaged over 3 independent runs. When inte-

Table 2: UDG performances on DomainNet subset. Best methods are highlighted in **bold**.

| Sources | painting ∪ real ∪ sketch | | | clipart ∪ infograph ∪ quickdraw | | | | |
|---|---|---|---|---|---|---|---|---|
| Target | clipart | infograph | quickdraw | painting | real | sketch | Overall | Avg. |
| Label Fraction 1% | | | | | | | | |
| ERM | 6.54 | 2.96 | 5.00 | 6.68 | 6.97 | 7.25 | 5.88 | 5.90 |
| BYOL | 6.21 | 3.48 | 4.27 | 5.00 | 8.47 | 4.42 | 5.61 | 5.31 |
| MoCo V2 | 18.85 | 10.57 | 6.32 | 11.38 | 14.97 | 15.28 | 12.12 | 12.90 |
| AdCo | 16.16 | 12.26 | 5.65 | 11.13 | 16.53 | 17.19 | 12.47 | 13.15 |
| SimCLR V2 | 23.51 | 15.42 | 5.29 | 20.25 | 17.84 | 18.85 | 15.46 | 16.86 |
| DARLING | 18.53 | 10.62 | 12.65 | 14.45 | 21.68 | 21.30 | 16.56 | 16.54 |
| DiMAE | 26.52 | 15.47 | 15.47 | 20.18 | 30.77 | 20.03 | 21.85 | 21.41 |
| BrAD | 47.26 | 16.89 | 23.74 | 20.03 | 25.08 | 31.67 | 25.85 | 27.45 |
| CycleMAE | 37.54 | 18.01 | 17.13 | 22.85 | 30.38 | 22.31 | 24.08 | 24.71 |
| SimCLR w/ FA | 60.83 | 18.42 | 26.31 | 24.29 | 29.73 | 40.29 | 30.82 | 33.31 |
| SimCLR w/ BSS | **61.94** (↑ 1.11) | 19.58 (↑ 1.16) | **26.98** (↑ 0.67) | 27.40 (↑ 3.11) | 31.55 (↑ 1.82) | 41.49 (↑ 1.2) | 32.27 (↑ 1.45) | 34.82 (↑ 1.51) |
| SWaV w/ FA | 59.27 | **20.95** | 18.94 | 30.99 | 35.73 | 45.28 | 31.87 | 35.19 |
| SWaV w/ BSS | 60.40 (↑ 1.13) | 20.12 (↓ 0.83) | 23.09 (↑ 4.15) | **34.64** (↑ 3.65) | **38.45** (↑ 2.72) | **46.90** (↑ 1.62) | **34.32** (↑ 2.45) | **37.27** (↑ 2.08) |
| Label Fraction 5% | | | | | | | | |
| ERM | 10.21 | 7.08 | 5.34 | 7.45 | 6.08 | 5.00 | 6.50 | 6.86 |
| BYOL | 9.60 | 5.09 | 6.02 | 9.78 | 10.73 | 3.97 | 7.83 | 7.53 |
| MoCo V2 | 28.13 | 13.79 | 9.67 | 20.80 | 24.91 | 21.44 | 18.99 | 19.79 |
| AdCo | 30.77 | 18.65 | 7.75 | 19.97 | 24.31 | 24.19 | 19.42 | 20.94 |
| SimCLR V2 | 34.03 | 17.17 | 10.88 | 21.35 | 24.34 | 27.46 | 20.89 | 22.54 |
| DARLING | 39.32 | 19.09 | 10.50 | 21.09 | 30.51 | 28.49 | 23.31 | 24.83 |
| DiMAE | 42.31 | 18.87 | 15.00 | 27.02 | 39.92 | 26.50 | 27.85 | 28.27 |
| BrAD | 64.01 | **25.02** | 29.64 | 29.32 | 34.95 | 44.09 | 35.37 | 37.84 |
| CycleMAE | 55.14 | 20.87 | 19.62 | 27.64 | 40.24 | 28.71 | 30.80 | 32.04 |
| SimCLR w/ FA | 69.04 | 20.31 | 29.76 | 36.44 | 41.95 | 51.05 | 38.60 | 41.42 |
| SimCLR w/ BSS | **71.21** (↑ 2.17) | 20.93 (↑ 0.62) | **32.42** (↑ 2.66) | 36.68 (↑ 0.24) | 41.49 (↓ 0.46) | 52.75 (↑ 1.7) | 39.73 (↑ 1.13) | 42.58 (↑ 1.16) |
| SWaV w/ FA | 68.84 | 24.05 | 26.06 | 43.97 | 49.11 | 59.16 | 41.68 | 45.20 |
| SWaV w/ BSS | 70.56 (↑ 1.72) | 24.35 (↑ 0.3) | 28.83 (↑ 2.77) | **46.17** (↑ 2.2) | **51.21** (↑ 2.1) | **59.71** (↑ 0.55) | **43.53** (↑ 1.85) | **46.81** (↑ 1.61) |
| Label Fraction 10% | | | | | | | | |
| ERM | 15.10 | 9.39 | 7.11 | 9.90 | 9.19 | 5.12 | 8.94 | 9.30 |
| BYOL | 14.55 | 8.71 | 5.95 | 9.50 | 10.38 | 4.45 | 8.69 | 8.92 |
| MoCo V2 | 32.46 | 18.54 | 8.05 | 25.35 | 29.91 | 23.71 | 21.87 | 23.00 |
| AdCo | 32.25 | 17.96 | 11.56 | 23.35 | 29.98 | 27.57 | 22.79 | 23.78 |
| SimCLR V2 | 37.11 | 19.87 | 12.33 | 24.01 | 30.17 | 31.58 | 24.28 | 25.84 |
| DARLING | 35.15 | 20.88 | 15.69 | 25.90 | 33.29 | 30.77 | 26.09 | 26.95 |
| DiMAE (full fine-tune) | 70.78 | 38.06 | 27.39 | 50.73 | 64.89 | 55.41 | 49.49 | 51.21 |
| BrAD | 68.27 | 26.60 | **34.03** | 31.08 | 38.48 | 48.17 | 38.74 | 41.10 |
| CycleMAE(full fine-tune) | **74.87** | **38.42** | 28.32 | **52.81** | **67.13** | 56.37 | **50.78** | **52.98** |
| SimCLR w/ FA | 70.12 | 20.50 | 31.23 | 39.16 | 44.45 | 52.87 | 40.31 | 43.05 |
| SimCLR w/ BSS | 71.95 (↑ 1.83) | 21.27 (↑ 0.77) | 33.47 (↑ 2.24) | 39.49 (↑ 0.33) | 44.67 (↑ 0.22) | 55.42 (↑ 2.55) | 41.57 (↑ 1.26) | 44.38 (↑ 1.33) |
| SWaV w/ FA | 69.81 | 24.39 | 28.97 | 45.92 | 50.79 | **60.78** | 43.46 | 46.78 |
| SWaV w/ BSS | 71.99 (↑ 2.18) | 24.34 (↓ 0.05) | 29.82 (↑ 0.85) | 48.28 (↑ 2.36) | 52.37 (↑ 1.58) | 60.55 (↓ 0.23) | 44.59 (↑ 1.13) | 47.89 (↑ 1.11) |

grating BSS, almost all target domain accuracies increase while per-domain-averaged accuracy always improves: SimCLR → (+1.51%, +1.16%, +1.33%); SWaV → (+2.08%, +1.61%, +1.11%) for the labeled data fractions 1%, 5% and 10%, respectively.

**Camelyon17 WILDS.** Averaged accuracy over 10 independent runs, for different fractions of labeled data, on the `test` split are reported on Table 3. To ensure fair comparisons, the reimplemented methods DARLING and DiMAE used identical pretraining and fine-tuning hyperparameters as extended SSL methods. For each extended SSL method, BSS leads to substantial performance gains

Table 3: UDG performances on Camelyon17 WILDS. Best methods are highlighted in **bold**.

| | Label Fraction | | | |
|---|---|---|---|---|
| Method | 1% | 5% | 10% | 100% |
| DARLING[*] | 70.44 | 72.00 | 72.43 | 72.36 |
| DiMAE[*] | 89.81 | 89.17 | 89.77 | 90.40 |
| FixMatch[†] | ✗ | ✗ | ✗ | 71.00 |
| SWaV[†] | ✗ | ✗ | ✗ | 91.40 |
| SimCLR w/ FA | 90.82 | 93.00 | 92.94 | 93.44 |
| SimCLR w/ BSS | **92.27** (↑ 1.45) | **94.75** (↑ 1.75) | **94.82** (↑ 1.88) | **95.00** (↑ 1.56) |
| SWaV w/ FA | 91.03 | 91.94 | 91.96 | 92.26 |
| SWaV w/ BSS | **93.42** (↑ 2.39) | **93.99** (↑ 2.05) | **94.07** (↑ 2.11) | **94.08** (↑ 1.82) |
| MSN w/ FA | 87.07 | 87.33 | 87.55 | 88.82 |
| MSN w/ BSS | **90.93** (↑ 3.86) | **91.82** (↑ 4.49) | **91.83** (↑ 4.28) | **91.98** (↑ 3.76) |

[*] Our implementation (no available public code).
[†] From WILDS challenge (Koh et al., 2021). Uses unlabeled data from the target domain.

ranging from $+1.45\%$ to $+4.49\%$ resulting in state-of-the-art performances. Extended SSL methods with BSS even surpass methods trained with additional unlabeled target data.

# 5 ABLATION STUDIES AND ADDITIONAL EXPERIMENTS

## 5.1 AUGMENTATION STRATEGY ABLATION STUDY

To evaluate the benefits of each component in our augmentation strategy (BSS + additional augmentations) on performances, we pretrained a model on Camelyon17 WILDS using SimCLR and different combinations of components. For each combination and each fraction of labeled data, averaged accuracy over 10 independent runs are reported in Table 4. Sample-wise color jittering and

Table 4: SimCLR's performances on Camelyon17 WILDS varying our augmentation strategy's components. Best methods are highlighted in **bold**.

| Color-jitter | FA | BSS | Label Fraction | | | |
|---|---|---|---|---|---|---|
| | | | 1% | 5% | 10% | 100% |
| Sample-wise | ✗ | ✗ | 63.54 | 63.58 | 63.50 | 65.06 |
| Batch-wise | ✗ | ✗ | 66.81 | 68.19 | 68.45 | 68.53 |
| Sample-wise | ✓ | ✗ | 90.82 | 93.00 | 92.94 | 93.44 |
| Batch-wise | ✗ | ✓ | **92.27** | **94.75** | **94.82** | **95.00** |

no FA or BSS, resulting in the regular SimCLR, leads to poor performance for the different fractions of labeled data. Changing the color-jittering from sample-wise to batch-wise slightly improves performances suggesting that even reducing the styles variability of augmented images helps for generalization. FA leads to drastic performance gains compared to the regular SimCLR which is not surprising given their prior success in DG tasks. Combining SimCLR with BSS and batch-wise color-jitter yields even greater performance improvements and non-negligible gains compared to FA. This observation is also supported by results from Tables 1, 2, 3.

## 5.2 UNDERLYING MECHANISMS INVOLVED IN BSS EFFICIENCY

**Spurious correlations reduction, harder negatives creation and reduced batch size requirement (SimCLR).** We hypothesized that BSS should help reduce the emergence of spurious correlations when repelling negatives from positives. Additionally, standardizing styles among positives and negatives should also facilitate the creation of harder negatives, a known factor contributing to robust performance (Kalantidis et al., 2020; Robinson et al., 2020), while also reducing the demand for large batch sizes. To validate these hypotheses, we conducted 3 comprehensive experiments employing SimCLR with standard augmentation, FA, or BSS on Camelyon17 WILDS: (1) To validate BSS's effectiveness in reducing spurious correlations, we computed the averaged domain purity for representations of unseen source and target examples after pretraining. This metric, which quantifies the degree to which each example and its nearest neighbors share the same domain label, serves as an indicator of the domain-invariance within SSL representations (refer to Figure 2a). (2) To assess the impact of BSS on encouraging the presence of harder negatives, we computed representations for several augmented batches and calculated cosine similarities across all possible (anchor, negative) pairs, reporting the values in a histogram (refer to Figure 2b). (3) Lastly, to assess BSS's ability to mitigate the demand for large batch sizes, we pretrained SimCLR varying batch sizes and assessed performance using linear probing (refer to Figure 2c). Figure 2a demonstrates that SimCLR with FA exhibits slightly lower average domain purity than standard SimCLR while SimCLR with BSS results in much lower average domain purity. This observation affirms BSS's effectiveness in attenuating spurious correlations and enhancing domain-invariance. Figure 2b indicates that standard SimCLR tends to produce negatives that are dissimilar to the anchors. SimCLR with FA produces negatives more similar to the anchors but not to the same extent as SimCLR with BSS which confirms BSS's role in creating harder negatives. Finally, Figure 2c reveals that standard SimCLR yields considerably lower performances compared to SimCLR with FA or with BSS. FA and BSS lead to improved performances for any batch size. As batch size increases, performances augment until a plateau is reached. However, when using BSS, this plateau is reached for a lower batch size supporting BSS's efficacy in reducing the need for large batches.

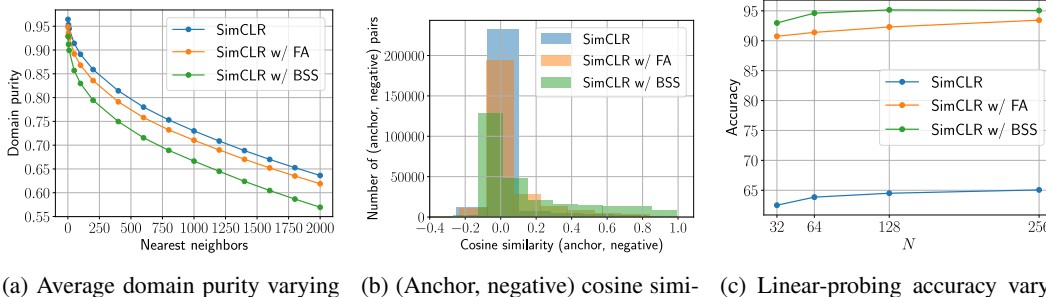

(a) Average domain purity varying number of nearest neighbors

(b) (Anchor, negative) cosine similarities

(c) Linear-probing accuracy varying batch size $N$

Figure 2: SimCLR experiments with standard augmentation, FA or BSS on Camelyon17 WILDS.

**Better domain heterogeneity and class homogeneity for examples assigned to the same prototype (SWaV, MSN).** We postulated that the coexistence of multiple domains/styles within views used for cluster assignments computation (i.e.: global views for SWaV and unmasked views for MSN) could introduce correlations between the assignments and domains/styles. To investigate the effectiveness of BSS in mitigating these correlations and shed light on why BSS yields superior representations compared to FA, we conducted the following experiment: We pretrained a backbone with SWaV using FA or BSS on DomainNet (sources: *painting* ∪ *real* ∪ *sketch*, targets: *clipart* ∪ *infograph* ∪ *quickdraw*). During training, at every $1K$ optimization step, we computed representations from unseen source and target examples along with their hard assignments resulting from SK. Subsequently, we evaluated the homogeneity of representations assigned to each prototype in terms of domain or class labels and averaged the homogeneity scores over all prototypes. The evolution of the averaged homogeneity score with respect to domain or class labels are respectively reported in Figure 3a and Figure 3b. Figure 3a reveals that employing BSS instead of

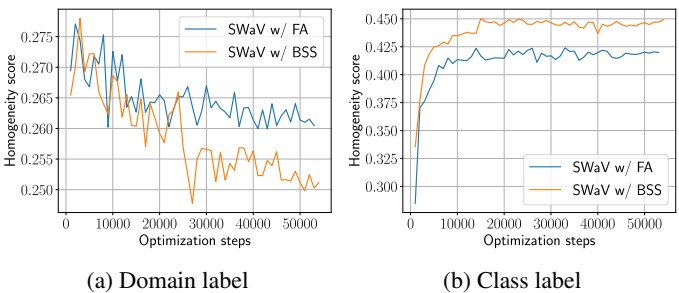

(a) Domain label

(b) Class label

Figure 3: Averaged homogeneity scores for representations assigned to the same prototype.

FA tends to reduce domain homogeneity (or improve domain heterogeneity) among representations assigned to the same prototype. This observation confirms BSS's role in reducing correlations between assignments and domains. Conversely, Figure 3b illustrates that BSS results in higher class label homogeneity attesting that BSS helps to produce assignments more semantically coherent.

## 6 CONCLUSION

This work introduces Batch Styles Standardization, an image style standardization technique to be combined with existing SSL methods to address UDG. Extending existing SSL methods with BSS offers serious advantages over prior UDG methods, including the elimination of domain labels and domain-specific network components dependencies to enhance domain-invariance while offering versatility for integration. Leveraging BSS, the extended SSL methods exhibit improved generalization capabilities, often surpassing or competing with alternative UDG strategies. Comprehensive experiments provide insights into the underlying mechanisms involved in BSS's efficiency. Other style transfer techniques, like GAN-based methods or AdaIN, could standardize image style to reduce spurious correlations in SSL. However, we leave this exploration for future research.

## 7 ACKNOWLEDGEMENTS

This project was provided with computer and storage resources by GENCI at IDRIS thanks to the grant 2022-AD011013424R1 on the supercomputer Jean Zay's V100 partition. This work was also partially supported by ANR-21-CE45-0007 (Hagnodice). We would like to express our sincere gratitude to Spyros Gidaris and Enzo Ferrante for their valuable feedback and comments, which greatly contributed to the improvement of this research.

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

## A   PSEUDO-CODE AND PYTORCH IMPLEMENTATION OF BATCH STYLES STANDARDIZATION

On Algorithm 1 and Listing 1, a pseudo-code along with a PyTorch implementation of Batch Styles Standardization are provided. The full code will be released upon acceptance.

---

**Algorithm 1:** Batch Styles Standardization

---

**Input** :
- $\{X_i\}_{1 \leq i \leq N}$: Batch of images
- $(r_{min}, r_{max})$: Minimum and maximum area ratios between the substituted amplitudes components and the full amplitude.

**Output:** Batch of images with standardized style: $\{\hat{X}_i\}_{1 \leq i \leq N}$

1   // Computes Fourier transform for all images in the batch
2   **for** $i \leftarrow 1$ **to** $N$ **do**
3     $\mathcal{A}(X_i) \leftarrow \sqrt{\mathrm{Re}\left(\mathcal{F}(X_i)\right)^2 + \mathrm{Im}\left(\mathcal{F}(X_i)\right)^2}$
4     $\mathcal{P}(X_i) \leftarrow \arctan\left(\dfrac{\mathrm{Im}\left(\mathcal{F}(X_i)\right)}{\mathrm{Re}\left(\mathcal{F}(X_i)\right)}\right)$
5   **end for**
6   // Sample the index $k$ associated to the style image and sample the amplitudes area ratio $r$
7   $k \sim U(\{1, \cdots, N\})$
8   $r \sim U(r_{min}, r_{max})$
9   // For each image, substitute the low-frequency components with those of the style image then apply the inverse Fourier transform to transfer the style onto the original image.
10 **for** $i \leftarrow 1$ **to** $N$ **do**
11     $\hat{\mathcal{A}}(X_i) \leftarrow \mathtt{substitute\_low\_freq}(\mathcal{A}(X_i), \mathcal{A}(X_k), r)$
12     $\hat{X}_i \leftarrow \mathcal{F}^{-1}\left(\hat{\mathcal{A}}(X_i)e^{-i\mathcal{P}(X_i)}\right)$
13 **end for**
14 **return** $\{\hat{X}_i\}_{1 \leq i \leq N}$

---

```python
class BatchStylesStandardization():
    """Implements Batch Styles Standardization. Given a
        batch of N images and their Fourier transforms,
        we manipulate the different amplitudes by
        substituting their low-frequency components
        with those a randomly chosen image.

    Attributes:
        ratios (tuple): $(r_min, r_max)$ specifying
            the minimum and maximum possible
            areas ratio between the substituted
            amplitude and the full amplitude.
    """
    def __init__(self, ratios):
        self.ratios = ratios

    def substitute_low_freq(self, src_amp, tgt_amp, ratio):
        """Substitute the low-frequency components
            of the source amplitudes with those
            of the target amplitudes.

        Args:
            src_amp (torch.Tensor): Source amplitudes
            tgt_amp (torch.Tensor): Target amplitudes
            ratio (float): Area ratio between the
                substituted amplitude and the full
                amplitude.

        Returns:
            torch.Tensor: Source amplitudes where
                the low-frequency components have been
                substituted with those
                of the target amplitudes.
        """
        # Compute center coordinates of amplitudes
        h, w = src_amp.shape[-2:]
        hc, wc = int(h//2), int(w//2)

        # Compute half length `l` of the components
        # to be substituted
        l = min([int(ratio*h/2), int(ratio*w/2)])

        # Substitute low freq components of source
        # amplitudes with those of the target amplitudes
        low_freq_tgt_amp = tgt_amp[
            ..., hc-l:hc+l, wc-l:wc+l]
        src_amp[
            ..., hc-l:hc+l, wc-l:wc+l] = low_freq_tgt_amp
        return src_amp

    def __call__(self, imgs, n_views):
        """Apply batch styles standardization `n_views` times
            on a batch of $N$ images.

        Args:
            imgs (torch.Tensor): Batch of images (N, 3, H, W)
            n_views (int): Number of augmented views

        Returns:
            torch.Tensor: Batch with standardized styles
                (N, n_views, 3, H, W)
        """
        # Apply FFT on source images
        fft = torch.fft.fftn(
            imgs, dim=(-2, -1))
        # Shift low-frequency components to the center
        fft = torch.fft.fftshift(
            fft, dim=(-2, -1))
        # Retrieve amplitude and phase
        amp, phase = fft.abs(), fft.angle()

        # Sample n_views images that will be used as
        # ref styles
        bs = imgs.size(0)
        sampled_ind = torch.randperm(bs)[:n_views]

        # Substitute low-freq of src amplitudes with those
        # of the n_views sampled images
        src_amp = amp.unsqueeze(1).repeat(
            [1, n_views, 1, 1, 1])
        tgt_amp = amp[sampled_ind].unsqueeze(0).expand(
            bs, -1, -1, -1, -1)
        sampled_ratio = random.uniform(*self.ratios)
        amp = self.substitute_low_freq(
            src_amp, tgt_amp, sampled_ratio)

        phase = phase.unsqueeze(1)
        # Reconstruct FFT from amp and phase
        fft = torch.polar(amp, phase)
        # Shift back low-frequency to their
        # original positions
        fft = torch.fft.ifftshift(fft, dim=(-2, -1))
        # Invert FFT
        imgs = torch.fft.ifftn(
            fft, dim=(-2, -1)).real.clamp(0, 1)
        return imgs
```

Listing 1: Batch Styles Standardization PyTorch implementation

## B  TECHNICAL DETAILS ABOUT SSL METHODS

### B.1  SIMCLR

SimCLR aims to bring representations of augmented views of the same image closer (positives) while repelling all other images representations (negatives). In practice, given a batch of $N$ images, each image is augmented $V$ times independently resulting in a $N \times V$ images grid where each row $c$ corresponds to a content and each column $s$ to a view. For each image $\boldsymbol{X}_{cs}$ and its corresponding representation $\boldsymbol{z}_{cs} \in \mathbb{R}^D$, SimCLR minimizes the NT-Xent loss with temperature $T$:

$$\mathcal{L}_{cs} = \frac{-1}{V-1} \sum_{s' \neq s} \log \left( \frac{e^{\boldsymbol{z}_{cs} \cdot \boldsymbol{z}_{cs'}/T}}{\displaystyle\sum_{(c'',s'') \neq (c,s)} e^{\boldsymbol{z}_{cs} \cdot \boldsymbol{z}_{c''s''}/T}} \right) \tag{6}$$

### B.2  SWAV

SWaV computes the representations of different views of the same image while clustering them using an online algorithm. Since representations should capture similar information, SWaV assumes that one view's cluster assignment can predicted from representations of other views. This swapped prediction idea is the core concept behind SWaV loss formulation.

Concretely, in SWaV, an image $\boldsymbol{X}_n$ is augmented into 2 views $\boldsymbol{X}_n^{(s)}$ and $\boldsymbol{X}_n^{(t)}$, with corresponding representations $\boldsymbol{z}_n^{(s)}$ and $\boldsymbol{z}_n^{(t)}$. Similarities between representations and $K$ learnable cluster centroids/prototypes $\boldsymbol{C} \in \mathbb{R}^{K \times D}$ are computed and converted into probabilities such as follows:

$$\boldsymbol{p}_n^{(v)} = \text{softmax} \left( \frac{\boldsymbol{z}_n^{(v)} \cdot \boldsymbol{C}^T}{\tau} \right) \quad \forall v \in \{s, t\} \tag{7}$$

To compute cluster assignments also referred to as codes and denoted $\boldsymbol{q}_n^{(v)}$, SWaV relies on the Sinkhorn-Klopp (SK) algorithm (Cuturi, 2013). SK is performed on all views representations trying to assign representations to the most similar centroids but also uniformly among clusters. Finally, based on the swapped prediction concept, SWaV minimizes the following per-sample loss:

$$\mathcal{L}_n = H(\boldsymbol{q}_n^{(s)}, \boldsymbol{p}_n^{(t)}) + H(\boldsymbol{q}_n^{(t)}, \boldsymbol{p}_n^{(s)}) \tag{8}$$

In Equation 8, $H(\boldsymbol{p}, \boldsymbol{q})$ stands for the cross-entropy between an approximated probability distribution $\boldsymbol{q}$ and a true probability distribution $\boldsymbol{q}$.

In practice, SWaV employs a multi-crop strategy, generating 2 global views (large crops) and $V$ local views (small crops) for each image. Cluster assignments are then computed only from the 2 global views while probabilities are derived from all the $V + 2$ views. In this setting, SWaV minimizes the following loss:

$$\mathcal{L}_n = \frac{1}{2(V+1)} \sum_{i=1}^{2} \sum_{v=1}^{V+2} \mathbf{1}_{i \neq v} H(\boldsymbol{q}_n^{(i)}, \boldsymbol{p}_n^{(v)}) \tag{9}$$

### B.3  MSN

Given two views of the same image, MSN randomly masks the patches of one view and leaves the other unchanged. Then, MSN's goal is to match the representation of the masked view with that of the unmasked view.

To derive a view's representation, MSN computes the similarities between its embedding and a set of cluster centroids/prototypes, subsequently transforming them into a probability distribution. As direct matching of these representations can lead to representation collapse, MSN simultaneously optimizes a cross-entropy term along with an entropy regularization term on the mean representation of the masked views. The entropy regularization term encourages the model to use the entire set of centroids/prototypes. Additionally, MSN employs Sinkhorn-Klopp on the representations of the unmasked views to avoid tuning the hyperparameter weighting the entropy regularization term.

In practice and more formally, MSN generates for each image $X_n$, $M$ masked views $\{X_{n,1}, \ldots, X_{n,M}\}$ and a single unmasked view $X_n^+$. Masked views are processed by a student encoder and the unmasked view by a teacher encoder whose weights are updated via an exponential moving average of the student encoder's weights. Masked and unmasked views' embeddings denoted $\{z_{n,1}, \ldots, z_{n,M}\}, z_n^+$ are then compared to a set of centroids/prototypes $C \in \mathbb{R}^{K \times D}$ and the resulting similarities are converted into probability distributions $\{p_{n,1}, \ldots, p_{n,M}\}, p_n^+$:

$$
\begin{cases}
p_{n,m} = \text{softmax}\left(\dfrac{z_{n,m} \cdot C^T}{\tau}\right) \\
p_n^+ = \text{softmax}\left(\dfrac{z_n^+ \cdot C^T}{\tau^+}\right)
\end{cases}
\tag{10}
$$

$\tau$ and $\tau^+$ stand for temperature hyperparameters and are chosen such that $\tau > \tau^+$ to encourage sharper probability distributions implicitly guiding the model to produce confident masked views representations. Given a batch of $N$ images, MSN minimizes the following loss:

$$
\begin{cases}
\mathcal{L} = \dfrac{1}{NM} \sum_{n=1}^{N} \sum_{m=1}^{M} H(p_n^+, p_{n,m}) - \lambda H(\bar{p}) \\
\bar{p} = \dfrac{1}{NM} \sum_{n=1}^{N} \sum_{m=1}^{M} p_{n,m}
\end{cases}
\tag{11}
$$

$H(p, q)$ stands for the cross-entropy between an approximated probability distribution $q$ and a true probability distribution $q$ while $H(\bar{p})$ denotes the entropy of the masked views' mean representation $\bar{p}$.

## C  IMPLEMENTATION DETAILS

### C.1  PRETRAINING

On **PACS** and **DomainNet**, as part of the geometric augmentations, we use random crop resizing, horizontal flips, small rotations, cutout(DeVries & Taylor, 2017) while color augmentations are applied in batch-wise manner using color jitter, random equalize, random posterize, random solarize and random grayscale. On **Camelyon17 WILDS**, we use random crop resizing, flips, rotations, cutout and batch-wise color jitter. All other hyperparameters for SimCLR, SWaV, MSN are respectively specified on Tables 5, 6, 7.

Table 5: Hyperparameters used for SimCLR extension based on Batch Styles Standardization

| datasets | PACS | DomainNet | Camelyon17 WILDS |
|---|---|---|---|
| backbone | ResNet-18 | ResNet-18 | ResNet-50 |
| $V$ | $2 \times 224^2 + 6 \times 128^2$ | $2 \times 224^2 + 6 \times 128^2$ | $8 \times 128^2$ |
| $(r_{min}, r_{max})$ | $(0.02, 1)$ | $(0.02, 1)$ | $(0.02, 0.1)$ |
| $D$ | 128 | 128 | 128 |
| $T$ | 0.5 | 0.5 | 0.5 |
| $N$ | 256 | 512 | 256 |
| steps | 60K | 60K | 150K |
| optimizer | LARS | LARS | LARS |
| learning rate | 0.2 | 0.4 | 0.2 |
| learning rate schedule | linear warmp-up + cosine decay | linear warmp-up + cosine decay | linear warmp-up + cosine decay |
| weight decay | $10^{-6}$ | $10^{-6}$ | $10^{-6}$ |

### C.2  FINE-TUNING/LINEAR-PROBING

For all datasets (**PACS**, **DomainNet**, and **Camelyon17 WILDS**), we use the Adam optimization method (Kingma & Ba, 2014) with an initial learning rate of $10^{-4}$, a learning rate scheduler with cosine decay, and weight decay of $10^{-4}$. The networks are trained respectively for 5K, 1K, and 15K steps with batch sizes of 128, 64, and 64. When performing linear probing, we follow the same normalization scheme as (He et al., 2022) by adding a batch normalization layer (Ioffe & Szegedy, 2015) without affine parameters before the linear classifier.

Table 6: Hyperparameters used for SWaV extension based on Batch Styles Standardization

| datasets | PACS | DomainNet | Camelyon17 WILDS |
|---|---|---|---|
| backbone | ResNet-18 | ResNet-18 | ResNet-50 |
| global views | $2 \times 224^2$ | $2 \times 224^2$ | $2 \times 128^2$ |
| local views | $6 \times 128^2$ | $6 \times 128^2$ | $6 \times 128^2$ |
| $(r_{min}, r_{max})$ | $(0.02, 1)$ | $(0.02, 1)$ | $(0.02, 0.1)$ |
| $K$ | 256 | 256 | 256 |
| $D$ | 128 | 128 | 128 |
| $\tau$ | 0.1 | 0.1 | 0.1 |
| $N$ | 256 | 512 | 256 |
| steps | 60K | 60K | 150K |
| optimizer | LARS | LARS | LARS |
| learning rate | 0.2 | 0.4 | 0.2 |
| learning rate schedule | linear warmp-up + cosine decay | linear warmp-up + cosine decay | linear warmp-up + cosine decay |
| weight decay | $10^{-6}$ | $10^{-6}$ | $10^{-6}$ |

Table 7: Hyperparameters used for MSN extension based on Batch Styles Standardization

| | |
|---|---|
| backbone | ViT-S/8 |
| unmasked views | $2 \times 96^2$ |
| masked views | $10 \times 64^2$ |
| $(r_{min}, r_{max})$ | $(0.02, 0.1)$ |
| patch masking ratio | 0.3 |
| $K$ | 128 |
| $D$ | 384 |
| $\tau$ | 0.1 |
| $\tau^+$ | 0.025 |
| $N$ | 256 |
| steps | 150K |
| Optimizer | LARS |
| learning rate | 0.2 |
| learning rate schedule | linear warm-up + cosine decay |
| weight decay | $10^{-6}$ |
| EMA momentum | 0.995 |

## D    DISCUSSIONS & ADDITIONAL VISUALIZATIONS

### D.1    TRANSFER LEARNING IN DG/UDG

The usage of Transfer Learning in DG/UDG is common but we think it is misguided. Often the pretraining dataset, such as Imagenet, can include one or more of the target domains, *e.g.*, *photo* for PACS or *real* for DomainNet. When evaluating on these domains, it is not possible to know if performances result from the generalization ability of the DG/UDG methods or from the transfer learning. For new DG/UDG methods, it is hard not to follow the common practice because transfer learning unfairly boosts the results of previous works, and state-of-the-art performances are often seen as a prerequisite for paper acceptance. We have tried to limit the usage of transfer learning in our experiments and only used it for DomainNet.

To demonstrate that UDG methods can unfairly benefit from ImageNet transfer learning, we have trained SimCLR with BSS both with and without ImageNet transfer learning. The resulting performances are reported in Table 8. These results clearly show that ImageNet transfer learning leads to a substantial improvement in UDG performance across all label fractions and for nearly all target domains.

### D.2    FEATURES VISUALIZATION

To assess the quality of the SSL representations and their ability to generalize across domains, we display, in Figure 4, t-SNE plots of the backbone representations for SimCLR with BSS and competitors on Camelyon17 WILDS. DARLING representations tend to be domain-invariant as lots of examples from different domains are superimposed. However, this is also the case for many examples from different classes indicating potentially poor model generalization. In contrast, DiMAE

Table 8: Impact of Imagenet transfer learning on UDG performances for PACS. Best methods are highlighted in **bold**. ↑ and ↓ stand for absolute gain or loss of performances when using Imagenet transfer learning.

| Methods | Label Fraction: 1% | | | | | Label Fraction: 5% | | | | |
|---|---|---|---|---|---|---|---|---|---|---|
| | Target domain | | | | | Target domain | | | | |
| | *photo* | *art* | *cartoon* | *sketch* | avg. | *photo* | *art* | *cartoon* | *sketch* | avg. |
| DiMAE† | 48.86 | 31.73 | 25.83 | 32.50 | 34.73 | 50.00 | 41.25 | 34.40 | 38.00 | 40.91 |
| BrAD† | **61.81** | 33.57 | 43.47 | 36.37 | 43.80 | **65.22** | 41.35 | 50.88 | 50.68 | 52.03 |
| CycleMAE† | 52.63 | 35.53 | 35.53 | 34.85 | 39.82 | 63.24 | 39.96 | 42.15 | 36.35 | 45.43 |
| SimCLR w/ BSS | 43.31 | 38.96 | 48.61 | 48.76 | 44.91 | 58.16 | 46.37 | 55.69 | **65.63** | 56.40 |
| SimCLR w/ BSS† | 42.04 (↓ 1.27) | **42.84** (↑ 3.88) | **55.37** (↑ 6.76) | **53.41** (↑ 4.65) | **48.42** (↑ 3.51) | 63.59 (↑ 5.43) | **53.65** (↑ 7.28) | **62.06** (↑ 6.37) | 62.81 (↓ 2.82) | **60.53** (↑ 4.13) |
| Methods | Label Fraction: 10% | | | | | Label Fraction: 100% | | | | |
| | Target domain | | | | | Target domain | | | | |
| | *photo* | *art* | *cartoon* | *sketch* | avg. | *photo* | *art* | *cartoon* | *sketch* | avg. |
| DiMAE† | 77.87 | 59.77 | 57.72 | 39.25 | 58.65 | 78.99 | 63.23 | 59.44 | 55.89 | 64.39 |
| BrAD† | 72.17 | 44.20 | 50.01 | 55.66 | 55.51 | ✗ | ✗ | ✗ | ✗ | ✗ |
| CycleMAE† | **85.94** | **67.93** | 59.34 | 38.25 | **62.87** | **90.72** | **75.34** | 69.33 | 50.24 | 71.41 |
| SimCLR w/ BSS | 63.29 | 51.37 | 59.43 | **66.09** | 60.04 | 79.50 | 62.73 | 65.67 | 73.02 | 70.23 |
| SimCLR w/ BSS† | 66.11 (↑ 2.82) | 56.04 (↑ 4.67) | **64.56** (↑ 5.13) | 61.86 (↓ 4.23) | 62.15 (↑ 2.11) | 85.35 (↑ 5.85) | 69.47 (↑ 6.74) | **73.36** (↑ 7.69) | **78.57** (↑ 5.55) | **76.69** (↑ 6.46) |

† Uses Imagenet transfer learning.

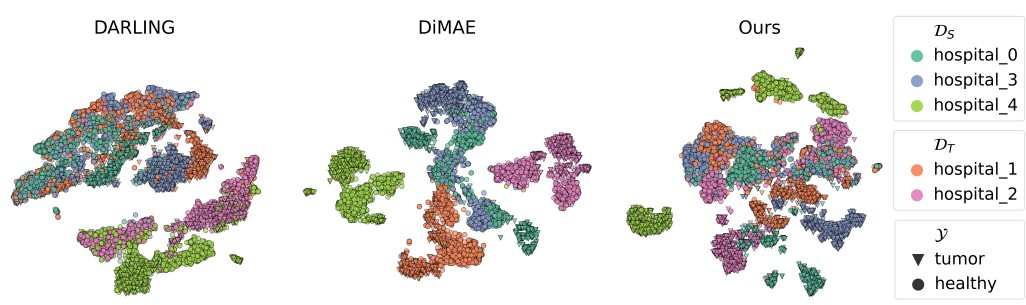

Figure 4: t-SNE plots of the backbone representations for different UDG methods on Camelyon17 WILDS. Colors and markers correspond respectively to different domains and classes. On the target domains (hospital_1, hospital_2), our method (SimCLR w/ BSS) shows better domain confusion while keeping better class separability. Zoom on pdf for better visualization.

representations appear to be well separated by classes but also by domains, especially for the target domains hospital_1 and hospital_2, indicating a lack of domain-invariance. Finally, better class separability and domain confusion emerge from the representations of SimCLR with BSS revealing a better domain-invariance and a potentially better cross-domain generalization.

### D.3 IMPACT OF $r$ ON BSS GENERATED IMAGES

To illustrate the effect of the hyperparameter $r$ on the generated images by BSS, we apply BSS on a single batch fixing the chosen style image and varying $r$. The resulting images are reported in Figure 5. We can observe that as $r$ increases, textures/styles with higher frequencies are transferred to the resulting images.

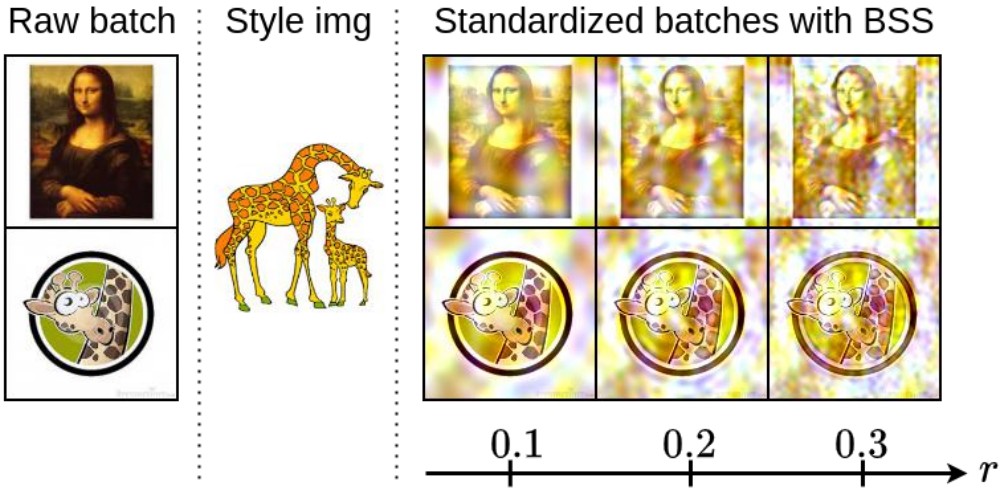

Figure 5: Impact of hyperparameter $r$ on augmented images with BSS

