# OpenReview forum: "Towards domain-invariant Self-Supervised Learning with Batch Styles Standardization"
_ICLR.cc/2024/Conference — ICLR 2024 poster_

### Official Review · Reviewer_qeEH · 2023-10-17

**Soundness:** 4 excellent
**Presentation:** 3 good
**Contribution:** 3 good
**Rating:** 6
**Confidence:** 4

**Summary:**

Existing UDG ( Unsupervised Domain Generalization) methods usually require samples to have domain labels for better learning of domain invariant features. The collection of domain labels is also costly in practical scenarios, which limits existing UDG. This paper proposes a BSS ( Batch Styles Standardization) approach in combination with existing SSL ( Self Supervised Learning ) methods, eliminating the need for domain labels. The authors combine the proposed BSS method with several SSL methods, experiment on UDG datasets, and obtain significant improvement in results.

**Strengths:**

1. This paper studies a novel problem, i.e. Unsupervised Domain Generalization without domain label. The basic idea is sound and very worthwhile.
2. Compared with the existing UDG methods, the experimental accuracy of the author's method has a significant advantage.
3. The writing is good, and the structure is easy to follow.
4. The ablation experiment was adequate.
5. In the part of comparative experiment, the author combined BSS with various types of SSL methods to demonstrate the universality of this method.

**Weaknesses:**

1. Inadequate ablation experiments.

In Section 5.2, the authors do not show the ablation effect of the original SimCLR. Table 4 demonstrates the effectiveness of FA and BSS compared to baseline SimCLR. However, it is not reflected in Section 5.2. So it does not show that FA and BSS are effective compared to
original SimCLR.

2. Perhaps there is a limitation to BSS.

Assuming that domain labels are not available, there is no guarantee that a number of randomly selected images belong to different domains. As the author said, "Finally, after applying the inverse Fourier transform to the different modified Fourier transforms, the style of the randomly chosen image is transferred to all images, effectively standardizing/harmonizing the style."
In the worst case, it is possible that a number of randomly selected images all belong to the same domain (same style), and then their magnitude spectra may be similar. If the magnitude spectra of these images are still used to augment all of the images, it may indeed create harder negative samples (because of the similar magnitude spectra). However, it would also create the problem of more similarity between pairs of positive samples, potentially forcing the network to capture more similar style information than domain invariant information in the positive sample pairs.

**Questions:**

1. What does "spurious correlations" mean? Is it the possibility that in SSL, when repelling negative samples, the main basis for error may be the difference in style?
2. In section 4.2, "We did not use ImageNet transfer learning, except on DomainNet to allow fair comparisons with prior UDG works." Does it mean that this paper uses an ImageNet pre-trained model when conducting experiments on the DomainNet dataset?

---

> ### Author Response · Authors · 2023-11-16
>
> Thank you for your time and insightful feedback. You will find below a response to your concerns/comments/questions.
>
> > **In Section 5.2, the authors do not show the ablation effect of the original SimCLR.**
>
> We have conducted the same ablation experiments as detailed in Section 5.2 specifically for a SimCLR with standard image data augmentation. The results of these additional experiments have been included to the existing plots on Figure 2. In examining the experiment on average domain purity varying with the number of nearest neighbors (Figure 2a), standard SimCLR exhibits only slightly higher average domain purity than SimCLR with FA, but significantly higher than SimCLR with Batch Style Standardization (BSS) (higher domain purity is worse). This observation suggests that FA does not significantly enhance domain-invariance compared to BSS. Similarly, in Figure 2b, it is noticeable that standard SimCLR results in negatives dissimilar to the anchors. SimCLR with FA produces negatives more similar to the anchors but not to the same extent as SimCLR with BSS. Finally, in terms of performance concerning batch size, standard SimCLR yields considerably lower performances compared to SimCLR with FA or with BSS.
>
> > **Perhaps there is a limitation to BSS. Assuming that domain labels are not available, there is no guarantee that a number of randomly selected images belong to different domains. [...] it may indeed create harder negative samples. However, it would also create the problem of more similarity between pairs of positive samples, potentially forcing the network to capture more similar style information than domain invariant information in the positive sample pairs.**
>
> Indeed, in the worst-case scenario, we could sample a batch of $N$ examples from a single domain (same style). Applying BSS successively $V$ times on this batch would result in
> $V$ batches of examples, each with its own style (as depicted in Fig 1c). As you correctly mentioned, because all examples initially came from the same domain (same style), we should end up with a single style for all examples, creating harder negatives but simpler positives. In contrastive learning, the loss tries at the same time to bring the positive pairs closer and to drive away the negative ones: since in this pathological case, all examples have the same style, the style cannot be used to do both at the same time. It should also be noted that on top of the style standardization, geometric augmentation and batch-wise color augmentations are applied (see section 3.2.2, ablation in section 5.1, table 4). This mitigates the problem since the $V$ positives have undergone different color augmentations.
>
> > **What does "spurious correlations" mean?**
>
> In general, correlations or features are said to be spurious when they predict the target label without possessing a causal relationship with it. In the context of SSL, spurious features would be typically associated with non-causal attributes such as style (color, texture, etc.), while the target variable would be the ones related to the SSL task objectives. As an example, in SimCLR, when diverse styles coexist among the positive and negative examples, “style features” may be used to solve the contrastive task.
>
> > **In section 4.2, "We did not use ImageNet transfer learning, except on DomainNet to allow fair comparisons with prior UDG works." Does it mean that this paper uses an ImageNet pre-trained model when conducting experiments on the DomainNet dataset?**
>
>  Using Transfer Learning / pretrained models in DG/UDG is a common practice in the community but we think it is misguided. Often the pre-training dataset, like ImageNet for example, can include one or more of the target domains  (e.g: photo for PACS). We have tried to limit as much as possible the usage of transfer learning in our experiments and only used it for the DomainNet dataset.

---

> > ### Comment · Reviewer_qeEH · 2023-11-17
> > **The standard SimCLR is not included in Figure2.**
> >
> > If the author-uploaded PDF is accurate, the three subfigures in Figure 2 indeed only contain SimCLR w/ FA and SimCLR w/ BSS (without a third standard SimCLR or SimCLR w/o any). Intuitively, SimCLR w/o any should perform worse than both SimCLR w/o FA and SimCLR w/o BSS because it does not use any cross-domain data augmentation. However, for rigor, it's still necessary for the authors to present the baseline SimCLR results to demonstrate the effectiveness of methods like BSS or FA for UDG.

---

> > ### Comment · Reviewer_qeEH · 2023-11-17
> > **The worst-case scenario of BSS**
> >
> > Thanks to the authors for the answers to the question. In fact, the probability of the worst-case scenario occurring is very small. And even if it occurs it's hard to say what effect it would have in training, which is a difficult abstract question to answer, so I apologize for being too harsh in suggesting a BSS limitation.
> >
> > In addition, I think BSS can be optimized to change the strategy of choosing images for augmenting. An immature idea is to design a function f(-, -) that compares the similarity of the amplitude spectra between samples, and by comparing the amplitude spectra of all the samples in a mini-batch, the top-k samples that have the lowest similarity with the other samples will be selected as the "chosen image".

---

> > > ### Author Response · Authors · 2023-11-17
> > > **Re: The worst-case scenario of BSS**
> > >
> > > > **In addition, I think BSS can be optimized to change the strategy of choosing images for augmenting. An immature idea is to design a function f(-, -) that compares the similarity of the amplitude spectra between samples, and by comparing the amplitude spectra of all the samples in a mini-batch, the top-k samples that have the lowest similarity with the other samples will be selected as the "chosen image".**
> > >
> > > Thank you for this interesting direction. This, indeed could be a possible strategy to ensure that the $V$ normalized batches possess significantly different styles.

---

> ### Author Response · Authors · 2023-11-17
> **Re: The standard SimCLR is not included in Figure2.**
>
> > **If the author-uploaded PDF is accurate, the three subfigures in Figure 2 indeed only contain SimCLR w/ FA and SimCLR w/ BSS (without a third standard SimCLR or SimCLR w/o any). Intuitively, SimCLR w/o any should perform worse than both SimCLR w/o FA and SimCLR w/o BSS because it does not use any cross-domain data augmentation. However, for rigor, it's still necessary for the authors to present the baseline SimCLR results to demonstrate the effectiveness of methods like BSS or FA for UDG.**
>
> We do not understand why you cannot see the new experiments concerning standard SimCLR on Figure 2. Normally, we had updated Figure 2 on our revised version submitted on **November 16, 2023, at 11:11 UTC**, including the requested experiments related to the standard SimCLR (blue color on each of the subfigures). However, the figure’s caption was not updated. Just in case, we have reuploaded the new version of the paper (with a more detailed caption). Kindly inform us if you still cannot see these new experiments.
>
>
> As you correctly intuited, the SimCLR performs worse than the SimCLR with FA and SimCLR with BSS since it does not use cross-domain augmentation (FA) or normalization (BSS)  that help a lot to achieve out-of-domain generalization.

---

> ### Comment · Reviewer_qeEH · 2023-11-20
> **About the latest uploaded version**
>
> **I apologize for the misunderstanding in my previous responses.** I mistakenly believed that the ICLR review process restricts the uploading of new paper versions, which led me to not download the PDF again.
>
> After re-reading the new version of the paper, the authors show the results of Standard SimCLR. As stated in the Rebuttal, **the performance of SimCLR is worse than SimCLR w/ FA and SimCLR w/ BSS**. Overall, the authors addressed previous Weaknesses in the newly uploaded version.

---

### Official Review · Reviewer_PTUx · 2023-10-25

**Soundness:** 3 good
**Presentation:** 2 fair
**Contribution:** 2 fair
**Rating:** 6
**Confidence:** 3

**Summary:**

The paper studies the problem of Unsupervised Domain Generalization (UDG) and proposes Batch Styles Standardization (BSS) for contrastive-based pretraining. It is a Fourier-based method that aims to standardize the style of images in a batch. The method can be plugged into many existing methods easily and shows good performance improvement across multiple benchmarks

**Strengths:**

- The paper is well-written and easy to follow.

- The motivation is clear and the method is simple and neat.

- The reported performance improvement is significant over the previous methods.

**Weaknesses:**

My major concern is that some important baselines are missing:
- What is the performance of the ERM (empirical risk minimization) on those benchmarks? People have observed ERM being a very strong baseline when it comes to domain generalization settings [1, 2].
- As a small fraction of labeled data is always used, why not try some good semi-supervised learning methods such as FixMatch [3] or AdaMatch [4] (which also deals with domain shift)? And there is also contrastive-based semi-supervised learning such as CoMatch [5]. Since the goal is to improve the performance on unseen target domains, what is the advantage of using a framework of unsupervised pretraining + finetuning?
- A related remark would be: what is the SOTA methodology when it comes to domain shift? In practice, one may easily resort to some VLMs (e.g. CLIP and its variants) pre-trained on large-scale image-text pair data when facing domain shift problems. As CLIP models have shown very good performance on samples under distribution shift, especially in image classification, I wonder to what extent can the problem be solved by them already. I think it makes more sense to develop techniques on top of these strong baselines, as it is very likely the method or the performance improvement on small-scale datasets or non-SOTA models does not transfer/scale well. It would be much more interesting to see if the proposed BSS still holds the same performance gap on top of CLIP. That being said, this remark is not a criticism of the authors who follow the common practice. But it would be still great if the authors could share their thoughts on this point.

Minor:
- To be more self-contained, it would be great if the authors could also introduce the contrastive-based UDG methods before Section 3.2.2.

[1] In Search of Lost Domain Generalization, Ishaan Gulrajani et al., ICLR 2021
[2] OoD-Bench: Quantifying and Understanding Two Dimensions of Out-of-Distribution Generalization, Nanyang Ye et al., CVPR2022
[3] FixMatch: Simplifying Semi-Supervised Learning with Consistency and Confidence, Kihyuk Sohn et al., NeurIPS 2020
[4] AdaMatch: A Unified Approach to Semi-Supervised Learning and Domain Adaptation, David Berthelot et al., ICLR 2022
[5] CoMatch: Semi-supervised Learning with Contrastive Graph Regularization, Junnan Li et al., ICCV 201

**Questions:**

- What are the source domains for PACS pretraining?

---

> ### Author Response · Authors · 2023-11-16
>
> Thank you for your time and relevant feedback. You will find below a response to your concerns.
>
> > **What is the performance of the ERM (empirical risk minimization) on those benchmarks?**
>
> Performances of the ERM baseline have been included in Tables 1 and 2. In our setting, ERM has a low performance since it is the only method that does not  benefit from the SSL pretraining step while the amount of labeled data used for finetuning is too small to make the model generalize well.
>
>
> > **As a small fraction of labeled data is always used, why not try some good semi-supervised learning methods such as FixMatch [3] or AdaMatch [4] (which also deals with domain shift)?
> [...] what is the advantage of using a framework of unsupervised pretraining + fine tuning?**
>
> This is a fair question: Indeed, in practical scenarios aimed at producing a model with robust generalization to unseen domains, a dilemma can arise regarding two primary learning strategies:
> 1. Unsupervised Pretraining handling domain shift (UDG) + Finetuning
> 2. Semi-Supervised Learning, such as FixMatch, combined with Domain Generalization techniques to handle domain shift. (Note: AdaMatch works in a Unsupervised Domain Adaptation setting not Domain Generalization)
> In practice, these strategies are not mutually exclusive: Combining both approaches would allow one to harness the strengths of each, potentially achieving optimal performance on unseen domains. Consequently, the suggested approach would be to (1) perform unsupervised pretraining addressing domain-shift (UDG) and subsequently (2) fine-tune the model using Semi-Supervised Learning in conjunction with Domain Generalization techniques.
>
> However, within the context of UDG, where the primary focus is evaluating the method's capacity to learn domain-invariant features, the typical workflow involves: (1) unsupervised training on source domains, (2) finetuning on source domains via linear probing, and (3) evaluation on unseen target domains. That is why the community follows this workflow.
>
> > **As CLIP models have shown very good performance on samples under distribution shift, especially in image classification, I wonder to what extent the problem can be solved by them already. [...]. It would be much more interesting to see if the proposed BSS still holds the same performance gap on top of CLIP [...] it would still be great if the authors could share their thoughts on this point.**
>
> It's indeed a very insightful remark. Vision-language models like CLIP have demonstrated impressive zero-shot capabilities in in-distribution evaluation scenarios and even in open-world evaluation scenarios. We acknowledge that VLMs have made strides in solving domain-shift problems and may serve as a strong foundation for developing new techniques for out-of-distribution (OOD) generalization.
>
> However, there are certain considerations and potential challenges when relying solely on VLMs for domain-shift problems:
>
> 1. VLMs often necessitate a substantial amount of labeled data for pretraining, and pretrained image and text encoders might not be well-suited for specific task data. For instance, in the histopathology field, existing pretrained VLM like CLIP does not generalize well while histopathology available paired data (images, diagnoses) are too scarce to train a new VLM from scratch [1].
> 2. While fine-tuning these models can yield significant performance gains on a given data distribution, it has been observed to potentially diminish out-of-distribution generalization ability [2, 3]. Hence, careful fine-tuning is essential to preserve their generalization capability in OOD evaluation scenarios.
> 3. On top of that, fair evaluation of their OOD capabilities are hard because VLMs are often trained on very large datasets that include many unknown domains. This makes it difficult to be sure that the target domains have not been seen during training.
>
> As suggested, it would be interesting to investigate whether BSS can enhance domain-invariance and improve performances on top of VLMs, thereby addressing the challenge of loss of OOD generalization during fine-tuning.
>
> References:
> * [1] Lu et al. "Towards a visual-language foundation model for computational pathology.", arXiv preprint 2023.
> * [2] Wortsman et al. "Robust fine-tuning of zero-shot models.", CVPR 2022
> * [3] Goyal, Sachin, et al. "Finetune like you pretrain: Improved finetuning of zero-shot vision models.”, CVPR 2023.
>
> >  **To be more self-contained, it would be great if the authors could also introduce the contrastive-based UDG methods before Section 3.2.2.**
>
> Contrastive-based UDG methods have been introduced before Section 3.2.2 (related works about UDG).
>
> > **What are the source domains for PACS pretraining?**
>
> For the experiments on PACS, we followed the common practice in the UDG community employing a leave-one-out domain evaluation. In Table 1, we only specifies the target domain for evaluation which implies that all other domains serve as pretraining.

---

> > ### Comment · Reviewer_PTUx · 2023-11-22
> > **Post-rebuttal decision**
> >
> > I thank the authors for their detailed response and additional experimental results. They address my concerns. Thus, I increased my score.

---

### Official Review · Reviewer_gWCx · 2023-10-27

**Soundness:** 2 fair
**Presentation:** 2 fair
**Contribution:** 2 fair
**Rating:** 5
**Confidence:** 5

**Summary:**

This paper studies the unsupervised domain generalization problem where there is a labeled training set, an unlabeled training set, and a test set. The main claim is that the style information should be standardized in training, which motivates the authors to propose BSS (Batch Style Standardization) to combine with self-supervised learning methods. Experiments on several benchmark datasets show the effectiveness of the BSS method.

-----Post rebuttal

I increased my score from 3 to 5 since they addressed my concerns. However, I did not give a 6 since I still think the novelty is limited.

**Strengths:**

1. The proposed method is simple and useful in image datasets.
2. The combination of BSS with SSL is interesting.
3. The results show that the BSS approach is effective compared to other counterparts.

**Weaknesses:**

1. The major weakness is novelty. Given that Fourier-based methods have been extensively studied in existing DG literatures, the direct adoption of Fourier features is not entirely novel. Plus, the idea of transforming the styles in batch is deeply related to Mixstyle [Zhou et al., ICLR21] but authors did not compare or discuss the difference.
2. The motivation of combining BSS with self-supervised learning is not clear. I can only see this: we can always combine them, that's all. I do not see the insights of such combination.
3. Section 3.2, i.e., the BSS part, is hard to understand. Authors should do their best to better present this part.
4. Comparision approaches are not enough: authors should compare with existing Fourier methods to validate their effectiveness.
5. There lacks theoretical support of why such BSS approach can succeed in learning domain-invariant representations.

**Questions:**

See weakness.

---

> ### Author Response · Authors · 2023-11-16
>
> Thank you for your time and constructive feedback. You will find below a response to your concerns.
>
> > **The idea of transforming the styles in a batch is deeply related to Mixstyle but authors did not compare or discuss the difference.**
>
> While MixStyle also transform the styles in the batch to improve out-of-domain generalization, significant differences exist between the proposed BSS and MixStyle:
> 1. The objectives of MixStyle and BSS diverge: MixStyle seeks to alter the style of examples, introducing a broad style variability at the features-level, whereas BSS strives to eliminate the style variability by enforcing a single style at the image-level (style augmentation vs style standardization).
> 2. To alter the style of examples at the features-level, MixStyle mixes the features maps` statistics (mean, std)  with those of random examples. In contrast, BSS normalizes the style of all examples in a batch by replacing all amplitudes with those of a randomly chosen image, resulting in a batch with a single style. In practice, MixStyle could be extended to normalize the style of examples by swapping all the statistics of a batch with those of a randomly chosen image and used within BSS to replace the proposed Fourier-based normalization.
> 3. MixStyle relies on the assumption that style information of examples can be captured by the examples’ statistics (mean, std) of bottom layers features in CNN [1]. This assumption is not directly transferable to all architecture like Vision Transformers for example. This makes MixStyle specific to the CNN architecture, whereas BSS operating at the image-level is architecture agnostic.
>
> References:
> * [1] Huang et al.. "Arbitrary style transfer in real-time with adaptive instance normalization.", ICCV 2017
>
> > **The motivation of combining BSS with self-supervised learning is not clear. [...]. I do not see the insights of such combination.**
>
> The primary motivation behind normalizing styles examples is to reinforce domain-invariance. Style normalization in a batch has the desired effect only if comparisons between examples of the batch occur during training. This is the case in the considered SSL methods but not in a standard supervised setting with a cross-entropy loss.
>
> The intuition behind why combining BSS to SSL methods helps to achieve domain- invariance is the following: both contrastive and non-contrastive SSL methods aim to distribute batch examples over an embedding space. This distribution over the embedding space can be driven explicitly by contrastive loss like in SimCLR or implicitly by methods preventing representation collapse like Sinkhorn-Knopp (SWaV, MSN), centering (DINO), or variance regularization (VicReg). However, when diverse styles coexist within a batch, there is no inherent mechanism preventing the distribution from being influenced by these different styles. In such case, these spurious correlations may harm the performance of the fine-tuned SSL model in an out-of-distribution evaluation scenario (UDG). Therefore, our hypothesis posits that normalizing the styles of examples in a batch should help reduce the emergence of spurious correlations in SSL features thereby enhancing domain-invariance.
>
> Empirical evidence supports this intuition, as substantial performance gaps between our method utilizing Fourier-based augmentations (FA) and BSS (styles normalization) have been observed (please refer to Table 1, 2, 3). Additionally,  the experiments provided in section 5.2 clarify the underlying mechanisms that contributed to BSS's effectiveness in improving domain-invariance in SSL representations.
>
> > **The BSS part, is hard to understand**
>
> We have tried to be as clear as possible by providing a description of the exact process of how BSS is performed (Section 3.2.2 - 2nd paragraph), a Figure illustrating how BSS is performed (Figure 1) and pseudo-code along with a Pytorch implementation of BSS. We will try to improve the explanation but it would help if you could explain which specific aspect you find unclear.
>
> > **Authors should compare with existing Fourier methods to validate their effectiveness.**
>
> All results from Table 1, 2 and 3 compare to existing UDG methods based on FA (DiMAE, CycleMAE (new)). Comparison to a standard FA strategy is also available for all experiments. If you can think of more pertinent comparisons, could you please point them out?
>
> > **Lacks theoretical support**
>
> In the UDG community, it is common practice to present motivations, intuitions, and empirical validations through performances or experiments (DARLING, BRaD, DiMAE, CycleMAE). Producing meaningful theoretical support for DG deep learning methods is hard:  strong assumptions regarding the data, such as a modelization of semantic/style separation, would be required and in practice, such assumptions never hold on real datasets. Moreover, our non-reliance on domain labels makes it even harder.

---

> > ### Comment · Reviewer_gWCx · 2023-11-23
> >
> > Thanks for the response. I carefully read the comments from other reviewers. Most of my concerns are addressed now. I increase my rating to 5. No further response needed.

---

### Official Review · Reviewer_eMsd · 2023-10-29

**Soundness:** 2 fair
**Presentation:** 3 good
**Contribution:** 2 fair
**Rating:** 6
**Confidence:** 4

**Summary:**

This paper proposes a batch-standardization method for domain-invariant self-supervised learning. The idea is borrowed mainly from Fourier domain adaptation. However, the new advantage is that it eliminates the requirements of domain labels. This paper validates the effectiveness on various benchmarks, such as PACS, DomainNet.

**Strengths:**

The biggest advantage of the proposed method is that it does not need domain labels to learn domain invariant features. It indeed reduces the requirements for domain labels. The experimental results show the effectiveness of the proposed BSS on various benchmarks.

**Weaknesses:**

[1] Originality: This paper proposes a batch-style standardization method to mix the domain styles in the batch. However, the idea is largely borrowed from Fourier Domain Adaptation [A], FACT[B] and Domain-invariant masked autoencoder [C]. The extension to samples in a mini-batch is also direct and does not need significant designs. Considering the author only claims one novelty, I do not think this paper is above the bar of ICLR.

[A] FDA: Fourier Domain Adaptation for Semantic Segmentation
[B] A Fourier-based Framework for Domain Generalization
[C] Domain Invariant Masked Autoencoders for Self-supervised Learning from Multi-domains


[2] The experimental results. I have also noticed that CycleMAE, which was published in the last ICLR, also lists the comparison of different pretrained models in Table 5. It shows comparable results with unsupervised learning pretrained models. In addition, the author should compare with CycleMAE [D].
[D] CYCLE-CONSISTENT MASKED AUTOENCODER FOR UNSUPERVISED DOMAIN GENERALIZATION

**Questions:**

Novelty and experiments are my most important concerns. Please carefully address my concerns listed in the weakness.

---

> ### Author Response · Authors · 2023-11-16
>
> Thank you for your time and constructive feedback. Your first point about novelty is treated in the global response. You will find below a response to your other remarks.
>
> > **The author should compare with CycleMAE**
>
> Thank you for pointing out this recent work that we missed during our initial submission. We have updated the related works accordingly and included performances of the method in Table 1 and 2.
>
> In examining the performances on PACS (Table 1), CycleMAE demonstrates slightly superior average performances at label fractions of 10% and 100%, compared to SimCLR with BSS. However, it should be noted that CycleMAE and our approaches differ in experimental settings:
> - CycleMAE employs ImageNet pretraining, while we do not.
> - Full finetuning is conducted on the 10% labeled data setting in CycleMAE, whereas we utilize linear probing as per BRaD experimental setting.
> - CycleMAE uses a ViT-small architecture, whereas most compared methods, including ours, utilize a ResNet18 architecture.
>
> Concerning the performances on the DomainNet subset (Table 2), our initial observations and comparisons remain consistent.

---

> > ### Comment · Reviewer_eMsd · 2023-11-23
> > **Thanks for your reply**
> >
> > Thanks for the reviewer's reply.
> >
> > I noticed that cycleMAE uses the ImageNet pretrain. However, I do not think it will lead to any problems because some pretraining methods directly related to the task also use ImageNet pretraining.
> >
> > Also, I recommend the author to try on ViT backbone since ViT has shown better scalability and modality-friendliness in recent years.
> >
> >
> > To conclude, I understand the author's response, and I think it is reasonable for the current paper, thus, I can slightly raise my score.
> >
> > However, I request the author to add results with ViT backbone and results using ImageNet pretraining.

---

> > > ### Author Response · Authors · 2023-11-23
> > >
> > > We thank the reviewer for this answer and for acknowledging our responses during the rebuttal. As requested, we are running experiments on PACS with a ViT backbone and with ImageNet pretraining. Given the insufficient remaining time for the rebuttal, we are unable to report these experiments now, but they will be included in the final version of the paper. Below, we are providing justifications regarding our experimental settings (architecture, pretraining):
> > >
> > > - **ResNet backbone**: On PACS and DomainNet, most UDG methods (except DiMAE and CycleMAE from the same authors) rely on ResNet architectures. This is why we have followed this experimental setting, relying on a convolutional backbone.
> > > - **ViT comparison on Camelyon17 WILDS**: We conducted experiments with a ViT backbone on the Camelyon17 WILDS dataset (Table 3 - MSN experiments). However, since CycleMAE does not provide evaluations on this dataset and because no code has been released by the authors, we cannot easily compare our results to theirs.
> > > - **Imagenet pretraining**: Using pretrained models in DG/UDG is a common practice in the community, but we strongly believe it can be misguided. Often, the pretraining dataset, like ImageNet, may include one or more of the target domains (e.g., photo for PACS). We have attempted to limit the usage of transfer learning in our experiments and discussed this choice in Appendix D.1. This is the reason why we have reported results on PACS without ImageNet pretraining, even if we compare with other methods that are using it (e.g. BraD, DiMAE, CycleMAE). However, we indeed expect better performance starting from a pretrained backbone and since this is requested from the reviewer, these experiments will be included.

---

### Official Review · Reviewer_qVbh · 2023-11-01

**Soundness:** 3 good
**Presentation:** 4 excellent
**Contribution:** 3 good
**Rating:** 6
**Confidence:** 2

**Summary:**

This paper proposes Batch Styles Standardization (BSS) to reduce spurious correlations in conventional self-supervised learning (SSL) representations, thereby making the resulting models generalize better on the test data drawn from an unseen domain. Specifically, the authors leverage the existing Fourier-based augmentation technique to transfer the style of a randomly chosen image to all other images within a batch. They also elaborate how BSS can be integrated with popular contrastive and non-contrastive SSL methods such as SimCLR, SWaV, and MSN. Experiments were conducted on 3 benchmark datasets for domain generalization to evaluate how BSS improves the performance of these SSL methods.

**Strengths:**

1. The paper is well written.
2. The authors perform a comprehensive literature review on SSL and unsupervised domain generalization.
3. The paper offers a clear explanation of how Fourier-based augmentation and BSS operate on images, making the methodology more reader-friendly for a broader audience.

**Weaknesses:**

For now I do not see any obvious weaknesses or technical flaws in the paper. However, it would be beneficial if the authors could provide further clarity on the novelty aspect. At the moment, it appears to be an application of Fourier-based augmentation to self-supervised learning.

Minor suggestions:
SimCLR and SWaV should be categorized as contrastive-based SSL methods in the second paragraph of Contributions.

**Questions:**

Can authors conduct some investigation on why BSS is sometimes outperformed (though by a small margin) by regular Fourier-based augmentation on the DomainNet dataset?

---

> ### Author Response · Authors · 2023-11-16
>
> Thank you for your time and constructive feedback. Your main concern about novelty is treated in the global response, your others remarks are covered below.
>
> > **SimCLR and SWaV should be categorized as contrastive-based SSL methods in the second paragraph of Contributions.**
>
> The categorization of SimCLR and SWaV as contrastive-based SSL methods have been clarified in an updated version of the paper in the second paragraph of Contributions.
>
> > **Investigation on why BSS is sometimes outperformed (though by a small margin) by regular Fourier-based augmentation on the DomainNet dataset?**
>
> We do not have a clear understanding of why BSS does not work as well for these few cases or a specific methodology to investigate them. However, it should be noted that, on average, BSS outperforms Fourier-based augmentation (FA) with a clear margin.

---

### Author Response · Authors · 2023-11-16

We sincerely thank all the reviewers for the time dedicated to our work and their thoughtful feedback that will help us improve the article.

We were pleased to read that reviewers found the paper **well-written** and **clear** (`qVbh`, `PTUx`, `qeEH`) and that they recognized the **effectiveness** and significant **performances improvements** (`eMsd`, `gWCx`, `PTUx`, `qeEH`) resulting from BSS. Moreover, the positive reception of the method's **simplicity** (`gWCx`, `PTUx`), **universality** (`qeEH`) and its ability to **eliminate the need for domain labels** (`eMsd`) is encouraging. However some concerns had been raised about the **novelty/originality of the method** (`qVbh`, `eMsd`, `gWCx`) and **additional comparisons/experiments** (`eMsd`, `gWCx`, `PTUx`, `qeEH`). In the following paragraphs, we address these two points but also provide a separate response for each reviewer.

**Novelty/originality of the method** (`qVbh`, `eMsd`, `gWCx`): Reviewers pointed out that Batch Style Standardization (BSS) appeared to be a "direct application" or "direct extension" (`qVbh`, `gWCx`, `eMsd`) of Fourier-based augmentation (FA) that “does not need significant designs” (`eMsd`) making the method not entirely novel.

We respectfully disagree with the lack of novelty/originality criticism. Our work aims to learn domain-invariant features in a Self-Supervised Learning setting (SSL) by eliminating the style variability in batch. Here are the following novelties/contributions of this work:

- **Reinforcing domain-invariance in SSL through style normalization, a novel approach**: The concept of reinforcing domain-invariance in SSL features through style standardization has not been previously explored and thereby is in-itself a novel way to approach the problem. We acknowledge that this contribution may not have been clearly communicated and consequently we will provide further clarification in the updated version of the paper.

- **BSS is not a direct application of FA**: To standardize the style and eliminate the style variability in batch, rather than swapping or mixing randomly the amplitudes among examples in batch as it is done in FA, BSS replace all amplitudes by those of a single randomly chosen image resulting in a batch of examples with a normalized style. While BSS is simple, which we consider a strength rather than a weakness, it is neither trivial nor a direct application of FA. Figures 1b and 1c visually depict the disparities at the batch level between FA (1b) and BSS (1c) applied to a batch of examples.  A direct application of FA (1b) yields a batch with diverse styles, whereas BSS (1c) produces a batch with a consistent style.

- **BSS and FA have different objectives**: Despite looking similar, BSS and FA objectives are fundamentally opposite: FA typically aims to alter the style of examples, introducing a broad style variability, whereas BSS strives to eliminate this variability by enforcing a single style.

- **Compared to prior UDG methods, BSS is versatile, eliminates the need for domain labels and has SOTA performance**: BSS versatility should be highlighted as it can be combined with several existing SSL methods to reinforce domain-invariance without relying on any domain labels or domain-specific architecture, a capability not achievable with previous Unsupervised Domain Generalization (UDG) approaches. Additionally, experiments on several UDG datasets have demonstrated that BSS combined with existing SSL methods significantly improves downstream task performances on unseen domains often outperforming or rivaling with UDG methods.

**Additional comparisons/experiments** (`eMsd`, `gWCx`, `PTUx`, `qeEH`):
1. `eMSD`: Comparisons with CycleMAE have been added to Table 1. and Table 2, with the reference included in the related work.
2. `gwCX` (“authors should compare with existing Fourier methods”): All results from Table 1, 2 and 3 already compare to existing UDG methods based on Fourier Augmentations (DiMAE, CycleMAE(new)). Additionally, to highlight the advantages of using BSS over Fourier-based augmentations (FA), all our method performances using either FA or BSS have already been reported to Table 1, 2 and 3.
3. `PTUx`: Empirical Risk Minimization (ERM) baselines have been included in Table 1, 2.
4. `qeEH` (the authors do not show the ablation effect of the original SimCLR): We have conducted the same ablation experiments as detailed in Section 5.2 specifically for a regular SimCLR. The results of these additional experiments have been included to the existing plots on Figure 2.

With these additional experiments, the conclusion and findings of the work remain the same.

All updates in the paper are highlighted in red for easy identification. We hope, through both global and individual responses, to have addressed the concerns raised. We would be happy to respond to further questions and suggestions if so.

---

### Meta-Review · Area_Chair_QxoF · 2023-12-02

**Metareview:**

This paper studies the problem of self-supervised learning (SSL). Specifically, the authors propose Batch Styles Standardization (BSS) based on Fourier augmentation to reduce spurious correlations, leads the models generalize better on the unseen testing data. The proposed method can be adopted to popular contrastive and non-contrastive SSL methods. Experiments on several datasets show the benefit of the proposed method.

**Strengths**

- The paper is well written and easy to follow.

- A comprehensive literature review is provided.

- Motivation of why using Fourier-based augmentation and applying BSS is well-presented.

- Extensive experiments are provided to demonstrate the benefit of the proposed method, where consistent improvements when applying to popular SSL methods.

**Weaknesses**

Most of the concerns about experiments and explanations are solved during rebuttal. However,

- The novelty of the proposed method is somewhat limited, which is agreed by most of the reviewers. The proposed method is mainly designed by widely used techniques, Fourier-based augmentation and transforming the styles in batch.

- Although the authors highlight the novelty during rebuttal, it is still not very convincing to the reviewers. Thus, the novelty and contribution statement should be further clarified and improved.

**Justification For Why Not Higher Score:**

The novelty of the proposed method is somewhat limited, it is mainly designed by widely used techniques, Fourier-based augmentation and transforming the styles in batch.

**Justification For Why Not Lower Score:**

The motivation of using Fourier-based augmentation and applying BSS for domain-invariant Self-Supervised Learning is clear and interesting. The proposed method can be adapted to different SSL methods and achieves consistent improvements. The experiments are extensive and solid.

---

### Decision · Program_Chairs · 2024-01-16

Accept (poster)